# Learning Gradient Boosted Decision Trees with Algorithmic Recourse

**Kentaro Kanamori**
Artificial Intelligence Laboratory
Fujitsu Limited
k.kanamori@fujitsu.com

**Ken Kobayashi**
School of Engineering
Institute of Science Tokyo
kobayashi.k@iee.eng.isct.ac.jp

**Takuya Takagi**
Artificial Intelligence Laboratory
Fujitsu Limited
takagi.takuya@fujitsu.com

## Abstract

This paper proposes a new algorithm for learning gradient boosted decision trees while ensuring the existence of recourse actions. Algorithmic recourse aims to provide a recourse action for altering the undesired prediction result given by a model. While existing studies often focus on extracting valid and executable actions from a given learned model, such reasonable actions do not always exist for models optimized solely for predictive accuracy. To address this issue, recent studies proposed a framework for learning a model while guaranteeing the existence of reasonable actions with high probability. However, these methods can not be applied to gradient boosted decision trees, which are renowned as one of the most popular models for tabular datasets. We propose an efficient gradient boosting algorithm that takes recourse guarantee into account, while maintaining the same time complexity as the standard ones. We also propose a post-processing method for refining a learned model under the constraint of a recourse guarantee and provide a PAC-style analysis of the refined model. Experimental results demonstrated that our method successfully provided reasonable actions to more instances than the baselines without significantly degrading accuracy and computational efficiency.

## 1 Introduction

Machine learning models are increasingly applied to critical decision-making tasks, such as loan approvals. In such high-stakes applications where predictions can significantly impact individuals [52], decision-makers need to explain how users can alter undesired predictions [42, 62]. *Algorithmic Recourse (AR)* aims to provide such information [58]. For a predictive model $f \colon \mathcal{X} \to \mathcal{Y}$, a desired class $y^* \in \mathcal{Y}$, and an instance $x \in \mathcal{X}$ such that $f(x) \neq y^*$, AR provides a perturbation $a$ that flips the prediction result into the desired class, i.e., $f(x + a) = y^*$. The user can regard the perturbation $a$ as a *recourse action* for obtaining the desired outcome $y^*$ [32]. For example, let us consider a situation where a bank deploys a model $f$ for predicting whether a loan applicant will repay the loan or default, and a user $x$ gets the loan application rejected. To help the user $x$ get the loan approved, AR suggests an action $a$ that changes the prediction result of $f$ from "default" to "repayment."

To provide actions that are executable for users, most of the existing studies on AR focus on post-hoc methods for extracting feasible actions with low costs from a learned model $f$ [5, 11, 13, 19, 27, 29, 39, 45, 48, 57, 64]. In general, however, such executable actions do not always exist when $f$ is trained solely for predictive performance [8, 36, 55]. For example, in the above loan approval

scenario, users can not change their demographic features (e.g., age or race) or features relating to past records (e.g., past bankruptcy). While users can change their education level, increasing it may be practically difficult due to the associated costs. If a learned model $f$ heavily relies on such features, there is no guarantee for the existence of executable actions, and thus, the existing post-hoc methods often fail to extract them from $f$. As a result, it may be impossible for many affected individuals to flip an undesired decision to their desired one as long as $f$ is deployed [56, 60].

To address the above issue, recent studies introduced a framework for learning a model $f$ that ensures the existence of executable actions $a$ for input instances $x$ with high probability [30, 51]. Specifically, Ross et al. [51] proposed a gradient descent algorithm for learning deep neural networks. Kanamori et al. [30] proposed a top-down greedy learning algorithm designed for classification trees. These studies demonstrate that we can learn a model $f$ that guarantees the existence of executable actions without significantly degrading predictive accuracy and computational efficiency [55].

In this paper, we focus on the *gradient boosted decision trees (GBDTs)* [16], such as XGBoost [7], LightGBM [33], and CatBoost [47]. Due to their performance and scalability, they are recognized as one of the state-of-the-art models for tabular datasets [18, 21, 31, 41], which often appear in the areas where AR is required (e.g., finance and justice) [61]. Contrary to their popularity and importance, however, existing methods for learning models with recourse guarantee cannot be applied to GBDTs. Ross et al. [51] assume that the loss function is differentiable with respect to the model parameters, which does not hold for tree-based models. The algorithm proposed by Kanamori et al. [30] is specifically tailored for classification trees and can not learn regression trees, which are used as base learners for GBDTs. In addition, although there exists gradient boosting algorithms considering some additional constraints [10, 22, 26, 46, 54, 59, 65], such as robustness [1, 4, 63], they do not aim to ensure the existence of recourse actions. Therefore, we need to design a new gradient boosting algorithm that takes the existence of recourse actions into account.

**Our contributions**    In this paper, we propose *Recourse-Aware gradient Boosted decIsion Trees (RA-BIT)*, a new framework for learning tree ensemble models that make accurate predictions and guarantee recourse actions. Our contributions are summarized as follows:

- We propose an efficient algorithm for learning tree ensembles by gradient boosting with the recourse loss [30, 51] that encourages the existence of recourse actions. To handle the recourse loss in the modern framework of gradient boosting, we derive its upper bound to which the Taylor expansion can be applied. We show that the computational complexity of our algorithm is equivalent to that of the standard gradient boosting algorithm.

- We introduce a post-processing task of modifying a learned tree ensemble model so as to satisfy the constraint on recourse guarantee. We formulate our task as a problem of refining the leaf weights of the trees in the ensemble, where we minimize the empirical risk under the constraint on the recourse loss. We show that this problem can be solved efficiently and provide a PAC-style guarantee for the models refined by our post-processing.

- We conducted numerical experiments on real datasets and demonstrated that our RABIT successfully provided executable recourse actions to more individuals than the baselines while keeping comparable predictive accuracy and computational efficiency. We also confirmed that RABIT could attain better trade-offs between predictive accuracy and recourse guarantee than the baselines by combining our learning algorithm and post-processing.

## 2    Problem statement

For a positive integer $n \in \mathbb{N}$, we write $[n] \coloneqq \{1, \ldots, n\}$. As with the previous studies [30, 51], we consider a binary classification task between undesired and desired classes. Note that our framework can be applied to a multiclass classification task if we can reduce it to a binary classification task between undesired and desired classes. Let $\mathcal{X} \subseteq \mathbb{R}^D$ and $\mathcal{Y} = \{\pm 1\}$ be input and output domains, respectively. We assume that $y = +1$ is a desirable class (e.g., loan repayment). We call a vector $\boldsymbol{x} = (x_1, \ldots, x_D) \in \mathcal{X}$ an *instance*, and a function $f \colon \mathcal{X} \to \mathbb{R}$ a *predictive model* that maps $\boldsymbol{x}$ to a prediction score. We assume that a predictive model $f$ makes a prediction $\hat{y} \in \mathcal{Y}$ according to $\hat{y} = \mathrm{sgn}(f(\boldsymbol{x}))$. Let $l \colon \mathcal{Y} \times \mathbb{R} \to \mathbb{R}_{\geq 0}$ be a differentiable convex loss function. We assume that a loss function $l$ satisfies $l_{01}(y, \hat{y}) \leq l(y, \hat{y})$ for the 0–1 loss function $l_{01}(y, \hat{y}) = \mathbb{I}[y \cdot \hat{y} < 0]$, and is non-increasing with respect to the value of $y \cdot \hat{y}$. These assumptions are satisfied by many major

loss functions, such as the binary cross-entropy loss. While these assumptions are required for our theoretical analyses, in practice, our learning algorithm works without these assumptions.

## 2.1 Algorithmic recourse

For an instance $x \in \mathcal{X}$, we define an *action* as a perturbation vector $a \in \mathbb{R}^D$ such that $x + a \in \mathcal{X}$. Let $\mathcal{A}(x)$ be a set of feasible actions for $x$ such that $\mathbf{0} \in \mathcal{A}(x)$ and $\mathcal{A}(x) \subseteq \{a \in \mathbb{R}^D \mid x + a \in \mathcal{X}\}$. For a model $f$, an action $a$ is *valid* for $x$ if $a \in \mathcal{A}(x)$ and $\mathrm{sgn}(f(x + a)) = +1$. For $x \in \mathcal{X}$ and $a \in \mathcal{A}(x)$, a *cost function* $c \colon \mathcal{A}(x) \to \mathbb{R}_{\geq 0}$ measures the required effort of $a$ with respect to $x$.

The aim of *Algorithmic Recourse (AR)* [58] is to find an action $a$ that is valid for $x$ with respect to $f$ and minimizes its cost $c(a \mid x)$. This task can be formulated as follows [32]:

$$\min_{a \in \mathcal{A}(x)} c(a \mid x) \quad \text{subject to} \quad \mathrm{sgn}(f(x + a)) = +1. \tag{1}$$

As with the existing studies [30, 51], we assume that the cost function $c$ satisfies the following properties: (i) $c(\mathbf{0} \mid x) = 0$; (ii) $c(a \mid x) = \max_{d \in [D]} c_d(a_d \mid x_d)$, where $c_d$ is the cost of the action $a_d$ for a feature $d$; (iii) $c_d(a_d \mid x_d) \leq c_d(a_d \cdot (1 + \varepsilon) \mid x_d)$ holds for any $\varepsilon \geq 0$. Note that several major cost functions, including the weighted $\ell_\infty$-norm [51] and max percentile shift [58], satisfy the above properties. We also assume that a set of feasible actions can be expressed as $\mathcal{A}(x) = [l_1, u_1] \times \cdots \times [l_D, u_D]$ with lower and upper bounds $l_d, u_d \in \mathbb{R}$ for $d \in [D]$. For example, while there exist some immutable features that can not be changed, such as age, race, and past bankruptcy, we can express them by setting $l_d = u_d = 0$. Similarly, we can express a feature that is allowed to be only increased (e.g., education level) by setting $l_d = 0$ and $u_d > 0$.

## 2.2 Tree ensemble and gradient boosting

A *tree ensemble model* $f$ is expressed as a sum of $T$ *regression trees* $f_1, \ldots, f_T \colon \mathcal{X} \to \mathbb{R}$; that is, $f(x) = \sum_{t=1}^T f_t(x)$. Each tree $f_t$ is a regressor that consists of a set of if-then-else rules expressed as a binary tree structure [3]. For a given input $x \in \mathcal{X}$, it makes a prediction according to the weight value $w$ of the leaf that $x$ reaches. The corresponding leaf is determined by traversing the tree from the root, depending on whether the *split condition* $x_d \leq b$ is true or not, where $(d, b) \in [D] \times \mathbb{R}$ is a pair of a feature and threshold of each internal node. Let $I_t$ be the total number of leaves in $f_t$, and $w_{t,i} \in \mathbb{R}$ be the *leaf weight* of a leaf $i \in [I_t]$, respectively. Then, we can express a regression tree $f_t$ as $f_t(x) = \sum_{i=1}^{I_t} w_{t,i} \cdot \phi_{t,i}(x)$, where $\phi_{t,i}(x) = \mathbb{I}[x \in r_{t,i}]$ is the leaf indicator of $i$ and $r_{t,i} \subseteq \mathcal{X}$ is the input region corresponding to $i$. Note that each region $r_{t,i}$ is an axis-aligned rectangle and determined by the split conditions on the path from the root to the leaf $i$ [15, 20].

*Gradient boosting* is one of the most popular frameworks for learning tree ensemble models [16, 18, 21, 41]. For each round $t \in [T]$, it learns a new regression tree $f_t$ that fits the pseudo residual corresponding to the loss $l(y, F_{t-1}(x))$ of the model $F_{t-1}(x) := \sum_{s=1}^{t-1} f_s(x)$ trained before round $t$ [9, 40]. Given a sample $S = \{(x_n, y_n)\}_{n=1}^N$, we consider the following optimization problem:

$$f_t^* \in \arg\min_{f_t \in \mathcal{F}} \sum_{n=1}^N l(y_n, F_{t-1}(x_n) + f_t(x_n)), \tag{2}$$

where $\mathcal{F}$ is a set of regression trees. For simplicity, we omit some popular techniques, such as regularization and learning rate [7, 16]. Note that our framework, proposed later, can handle them.

Because exactly solving the problem (2) is computationally challenging due to the combinatorial nature of decision trees [23, 24], existing methods employ a top-down greedy algorithm as with the standard decision tree learning [3]. Let us consider growing a regression tree $f_t$ by adding a new split condition $(d, b)$ to a leaf $i \in [I_t]$ of $f_t$. We assume a finite set of candidate thresholds $B_d \subset \mathbb{R}$ for each $d \in [D]$ such that $|B_d| = \mathcal{O}(N)$. Let $\mathcal{N}(r_{t,i}) = \{n \in [N] \mid x_n \in r_{t,i}\}$ be the set of instances that reach $i$, and $w_L, w_R \in \mathbb{R}$ be the leaf weights of the left and right children of $i$, respectively. Then, we consider the task of finding best parameters $(d, b, w_L, w_R)$, which can be formulated as

$$\min_{d \in [D], b \in B_d} \min_{w_L, w_R \in \mathbb{R}} \sum_{n \in \mathcal{N}(r_{t,i})} l(y_n, F_{t-1}(x_n) + h(x_n; d, b, w_L, w_R)), \tag{3}$$

where $h(x; d, b, w_L, w_R) := w_L \cdot \mathbb{I}[x_d \leq b] + w_R \cdot \mathbb{I}[x_d > b]$ is a decision stump with a split condition $(d, b)$ and leaf weights $w_L, w_R$. While the candidate split conditions $(d, b)$ are finite and can be easily enumerated, we need to numerically optimize the leaf weights $w_L, w_R$ for each $(d, b)$.

To efficiently solve the problem (3), modern implementations of the gradient boosted decision trees (GBDTs), such as XGBoost [7], employ the second-order Taylor expansion of the loss function $l$. Here, we fix the split condition $(d, b)$ and denote $r_{\mathrm{L}} \coloneqq \{\boldsymbol{x} \in r_{t,i} \mid x_d \le b\}$ and $r_{\mathrm{R}} \coloneqq r_{t,i} \setminus r_{\mathrm{L}}$. Let $\Phi_{d,b}(w_{\mathrm{L}}, w_{\mathrm{R}}) \coloneqq \sum_{n \in \mathcal{N}(r_{t,i})} l(y_n, F_{t-1}(\boldsymbol{x}_n) + h(\boldsymbol{x}_n; d, b, w_{\mathrm{L}}, w_{\mathrm{R}}))$ be the objective function of (3). By ignoring the constant term independent of $w_{\mathrm{L}}$ and $w_{\mathrm{R}}$, we can approximate $\Phi_{d,b}$ as follows [7]:

$$\Phi_{d,b}(w_{\mathrm{L}}, w_{\mathrm{R}}) \approx G_{\mathrm{L}} \cdot w_{\mathrm{L}} + \frac{1}{2} \cdot H_{\mathrm{L}} \cdot w_{\mathrm{L}}^2 + G_{\mathrm{R}} \cdot w_{\mathrm{R}} + \frac{1}{2} \cdot H_{\mathrm{R}} \cdot w_{\mathrm{R}}^2,$$

where $G_{\mathrm{L}} \coloneqq \sum_{n \in \mathcal{N}(r_{\mathrm{L}})} \frac{\partial}{\partial \hat{y}} l(y_n, \hat{y}) \mid_{\hat{y}=F_{t-1}(\boldsymbol{x}_n)}$ and $H_{\mathrm{L}} \coloneqq \sum_{n \in \mathcal{N}(r_{\mathrm{L}})} \frac{\partial^2}{\partial \hat{y}^2} l(y_n, \hat{y}) \mid_{\hat{y}=F_{t-1}(\boldsymbol{x}_n)}$. We define $G_{\mathrm{R}}$ and $H_{\mathrm{R}}$ in a similar way. We can compute the optimal leaf weights $w_{\mathrm{L}}^*$ and $w_{\mathrm{R}}^*$ that minimizes the above approximated objective function by $w_{\mathrm{L}}^* = -\frac{G_{\mathrm{L}}}{H_{\mathrm{L}}}$ and $w_{\mathrm{R}}^* = -\frac{G_{\mathrm{R}}}{H_{\mathrm{R}}}$, respectively. Thus, we can analytically obtain an approximate solution to the inner problem of (3) for each split condition $(d, b)$. While a naive computation of $w_{\mathrm{L}}^*$ and $w_{\mathrm{R}}^*$ requires $\mathcal{O}(N)$ time, we can compute them in amortized constant time if the instances $\boldsymbol{x}_n$ are sorted by $x_{n,d}$ in advance [7, 50]. Thus, we can obtain an approximate solution to the problem (3) in $\mathcal{O}(N \cdot D)$. We grow the $t$-th regression tree $f_t$ by recursively repeating this procedure until some conditions are met (e.g., maximum depth).

## 2.3 Problem formulation

To formulate our learning task, we introduce the *recourse loss* [30, 51]. For a cost budget parameter $\beta \ge 0$, let $\mathcal{A}_\beta(\boldsymbol{x}) \coloneqq \{\boldsymbol{a} \in \mathcal{A}(\boldsymbol{x}) \mid c(\boldsymbol{a} \mid \boldsymbol{x}) \le \beta\}$ be the set of feasible actions whose costs are lower than or equal to $\beta$. We define the recourse loss $l_\beta(\boldsymbol{x} \mid f)$ of a model $f$ for an instance $\boldsymbol{x}$ as follows:

$$l_\beta(\boldsymbol{x} \mid f) \coloneqq \min_{\boldsymbol{a} \in \mathcal{A}_\beta(\boldsymbol{x})} l(+1, f(\boldsymbol{x} + \boldsymbol{a}))$$

We can regard $l_\beta$ as a relaxation of the validity constraint with a cost budget $\beta$. By definition, it takes a small value if there exists a feasible action $\boldsymbol{a} \in \mathcal{A}(\boldsymbol{x})$ such that $\mathrm{sgn}(f(\boldsymbol{x} + \boldsymbol{a})) = +1$ and $c(\boldsymbol{a} \mid \boldsymbol{x}) \le \beta$. Thus, minimizing $l_\beta$ encourages $f$ to ensure a valid action $\boldsymbol{a}$ with a low cost [30, 51].

The aim of this paper is to propose a gradient boosting algorithm for learning a tree ensemble model $f$ while ensuring the existence of valid and low-cost recourse actions $\boldsymbol{a}$ for as many instances $\boldsymbol{x}$ as possible. More precisely, we learn a tree ensemble model $f(\boldsymbol{x}) = \sum_{t=1}^T f_t(\boldsymbol{x})$ that minimizes the weighted sum of the standard loss $l$ and recourse loss $l_\beta$. Given a sample $S = \{(\boldsymbol{x}_n, y_n)\}_{n=1}^N$, our learning task of each round $t \in [T]$ can be formulated as follows:

$$f_t^* \in \arg\min_{f_t \in \mathcal{F}} \sum_{n=1}^N \left( l(y_n, F_{t-1}(\boldsymbol{x}_n) + f_t(\boldsymbol{x}_n)) + \gamma \cdot l_\beta(\boldsymbol{x}_n \mid F_{t-1} + f_t) \right), \tag{4}$$

where $\gamma \ge 0$ is a hyperparameter that controls the trade-off between the predictive accuracy and recourse guarantee. We can recover the standard unconstrained gradient boosting by setting $\gamma = 0$.

## 3 Recourse-aware gradient boosting

In this section, we propose an efficient algorithm for learning a tree ensemble model while ensuring the existence of recourse actions for as many instances as possible. Following the modern frameworks of gradient boosting [7, 33, 47], we learn each regression tree $f_t$ by recursively solving the problem (4) in a top-down greedy manner and leveraging the second-order Taylor expansion to obtain a closed-form solution to the leaf weights $w_{\mathrm{L}}, w_{\mathrm{R}}$. Let $v_\beta(\boldsymbol{x}; r) \coloneqq \mathbb{I}[\exists \boldsymbol{a} \in \mathcal{A}_\beta(\boldsymbol{x}) : \boldsymbol{x} + \boldsymbol{a} \in r]$ be the indicator whether $\boldsymbol{x}$ can reach a region $r$ by some action $\boldsymbol{a} \in \mathcal{A}_\beta(\boldsymbol{x})$. Similar to the term $\Phi_{d,b}$ corresponding to the standard loss $l$, we define the term corresponding to the recourse loss $l_\beta$ as follows:

$$\Psi_{d,b}(w_{\mathrm{L}}, w_{\mathrm{R}}) \coloneqq \sum_{n \in \mathcal{N}_\beta(r_{t,i})} l_\beta(\boldsymbol{x}_n \mid F_{t-1} + h(\cdot; d, b, w_{\mathrm{L}}, w_{\mathrm{R}})),$$

where $\mathcal{N}_\beta(r_{t,i}) \coloneqq \{n \in [N] \mid v_\beta(\boldsymbol{x}_n; r_{t,i}) = 1\}$ is the set of instances $\boldsymbol{x}_n$ that can reach the leaf $i$ by some action $\boldsymbol{a} \in \mathcal{A}_\beta(\boldsymbol{x}_n)$. Then, we consider the following optimization problem:

$$\min_{d \in [D], b \in B_d} \min_{w_{\mathrm{L}}, w_{\mathrm{R}} \in \mathbb{R}} \Phi_{d,b}(w_{\mathrm{L}}, w_{\mathrm{R}}) + \gamma \cdot \Psi_{d,b}(w_{\mathrm{L}}, w_{\mathrm{R}}). \tag{5}$$

A main obstacle to solving the problem (5) is that we can not directly apply the second-order Taylor expansion to the term $\Psi_{d,b}$ due to the recourse loss $l_\beta$. Hence, efficiently solving the inner problem of (5) is not trivial in contrast to the standard case. To address this issue, we derive a differentiable upper bound of $\Psi_{d,b}$, which enables us to use the second-order Taylor expansion when solving the problem (5). We also show that we can compute the approximate solution to the problem (5) in $O(N \cdot D)$ time, which is equivalent to the standard unconstrained gradient boosting algorithm.

### 3.1 Differentiable upper bound on recourse loss

First, we derive an upper bound on the term $\Psi_{d,b}$ that is differentiable with respect to the leaf weights $w_{\mathrm{L}}$ and $w_{\mathrm{R}}$. In the following proposition, we show an upper bound on the recourse loss $l_\beta$.

**Proposition 1.** *For regression trees $f_1, \ldots, f_{t-1} \in \mathcal{F}$, we define $\xi_t(\boldsymbol{x}) := \sum_{s=1}^{t-1} \min_{\boldsymbol{a} \in \mathcal{A}_\beta(\boldsymbol{x})} f_s(\boldsymbol{x} + \boldsymbol{a})$. Then, for any instance $\boldsymbol{x} \in \mathcal{X}$ and regression tree $h \in \mathcal{F}$, we have*

$$l_\beta(\boldsymbol{x} \mid F_{t-1} + h) \leq \min_{\boldsymbol{a} \in \mathcal{A}_\beta(\boldsymbol{x})} l(+1, \xi_t(\boldsymbol{x}) + h(\boldsymbol{x} + \boldsymbol{a})).$$

We give a proof of Proposition 1 in Appendix A. Note that $\xi_t(\boldsymbol{x})$ is a lower bound on the prediction score of $F_{t-1}$ for $\boldsymbol{x}$ by some action $\boldsymbol{a} \in \mathcal{A}_\beta(\boldsymbol{x})$, and we can compute each term of $\xi_t(\boldsymbol{x})$ in $\mathcal{O}(I_s)$ [1]. Let $\bar{l}_\beta(w_{\mathrm{L}}, w_{\mathrm{R}}; \boldsymbol{x}) := \min_{\boldsymbol{a} \in \mathcal{A}_\beta(\boldsymbol{x})} l(+1, \xi_t(\boldsymbol{x}) + h(\boldsymbol{x} + \boldsymbol{a}; d, b, w_{\mathrm{L}}, w_{\mathrm{R}}))$ be the upper bound of Proposition 1 for a decision stump $h(\cdot; d, b, w_{\mathrm{L}}, w_{\mathrm{R}})$. For any $\boldsymbol{x}$ with $v_\beta(\boldsymbol{x}; r_{t,i}) = 1$, we have

$$\bar{l}_\beta(w_{\mathrm{L}}, w_{\mathrm{R}}; \boldsymbol{x}) = \begin{cases} \min_{w \in \{w_{\mathrm{L}}, w_{\mathrm{R}}\}} l(+1, \xi_t(\boldsymbol{x}) + w) & \text{if } v_{\mathrm{L}}(\boldsymbol{x}) = v_{\mathrm{R}}(\boldsymbol{x}) = 1, \\ l(+1, \xi_t(\boldsymbol{x}) + w_{\mathrm{L}} \cdot v_{\mathrm{L}}(\boldsymbol{x}) + w_{\mathrm{R}} \cdot v_{\mathrm{R}}(\boldsymbol{x})) & \text{otherwise,} \end{cases}$$

where $v_{\mathrm{L}}(\boldsymbol{x}) := v_\beta(\boldsymbol{x}; r_{\mathrm{L}})$ and $v_{\mathrm{R}}(\boldsymbol{x}) := v_\beta(\boldsymbol{x}; r_{\mathrm{R}})$. By definition, $v_\beta(\boldsymbol{x}; r_{t,i}) = 1$ implies $v_{\mathrm{L}}(\boldsymbol{x}) + v_{\mathrm{R}}(\boldsymbol{x}) \geq 1$. The case with $v_{\mathrm{L}}(\boldsymbol{x}) = v_{\mathrm{R}}(\boldsymbol{x}) = 1$ corresponds to the situation where the instance $\boldsymbol{x}$ can reach both left and right children by some action $\boldsymbol{a} \in \mathcal{A}_\beta(\boldsymbol{x})$, and the other case corresponds to the situation where only one of them is reachable. We can apply the Taylor expansion to the latter case, but not to the former, due to the minimum operator. To avoid this difficulty, we replace the minimum operator with the LogSumExp function as follows:

$$\tilde{l}_\beta(w_{\mathrm{L}}, w_{\mathrm{R}}; \boldsymbol{x}) := \begin{cases} \frac{\ln 2}{\nu} - \frac{1}{\nu} \ln \left( e^{-\nu \cdot l(+1, \xi_t(\boldsymbol{x}) + w_{\mathrm{L}})} + e^{-\nu \cdot l(+1, \xi_t(\boldsymbol{x}) + w_{\mathrm{R}})} \right) & \text{if } v_{\mathrm{L}}(\boldsymbol{x}) = v_{\mathrm{R}}(\boldsymbol{x}) = 1, \\ l(+1, \xi_t(\boldsymbol{x}) + w_{\mathrm{L}} \cdot v_{\mathrm{L}}(\boldsymbol{x}) + w_{\mathrm{R}} \cdot v_{\mathrm{R}}(\boldsymbol{x})) & \text{otherwise,} \end{cases}$$

where $\nu > 0$ is some small constant that controls the approximation quality. By LogSumExp trick, we have $\bar{l}_\beta(w_{\mathrm{L}}, w_{\mathrm{R}}; \boldsymbol{x}) \leq \tilde{l}_\beta(w_{\mathrm{L}}, w_{\mathrm{R}}; \boldsymbol{x})$. Combining the above results, we define our surrogate function of the recourse loss by $\tilde{\Psi}_{d,b}(w_{\mathrm{L}}, w_{\mathrm{R}}) := \sum_{n \in \mathcal{N}_\beta(r_{t,i})} \tilde{l}_\beta(w_{\mathrm{L}}, w_{\mathrm{R}}; \boldsymbol{x}_n)$. Our surrogate function $\tilde{\Psi}_{d,b}$ satisfies $\Psi_{d,b}(w_{\mathrm{L}}, w_{\mathrm{R}}) \leq \tilde{\Psi}_{d,b}(w_{\mathrm{L}}, w_{\mathrm{R}})$ and is differentiable with respect to $w_{\mathrm{L}}$ and $w_{\mathrm{R}}$ by definition.

### 3.2 Analytical solution to leaf weights

Now, we consider the following problem where we replace the term $\Psi_{d,b}$ of (5) with our surrogate $\tilde{\Psi}$:

$$\min_{d \in [D], b \in B_d} \min_{w_{\mathrm{L}}, w_{\mathrm{R}} \in \mathbb{R}} \Phi_{d,b}(w_{\mathrm{L}}, w_{\mathrm{R}}) + \gamma \cdot \tilde{\Psi}_{d,b}(w_{\mathrm{L}}, w_{\mathrm{R}}). \tag{6}$$

As with the modern gradient boosting algorithm [7, 33, 47], we obtain an approximate solution to the inner problem of (6) by applying the second-order Taylor expansion to its objective function. For notational convenience, we divide the set of instances $\mathcal{N}_\beta(r_{t,i})$ into three disjoint subsets:

$$\begin{aligned} \mathcal{N}_{\mathrm{B}} &:= \{n \in \mathcal{N}_\beta(r_{t,i}) \mid v_{\mathrm{L}}(\boldsymbol{x}_n) = 1, v_{\mathrm{R}}(\boldsymbol{x}_n) = 1\}, \\ \mathcal{N}_{\mathrm{L}} &:= \{n \in \mathcal{N}_\beta(r_{t,i}) \mid v_{\mathrm{L}}(\boldsymbol{x}_n) = 1, v_{\mathrm{R}}(\boldsymbol{x}_n) = 0\}, \\ \mathcal{N}_{\mathrm{R}} &:= \{n \in \mathcal{N}_\beta(r_{t,i}) \mid v_{\mathrm{L}}(\boldsymbol{x}_n) = 0, v_{\mathrm{R}}(\boldsymbol{x}_n) = 1\}. \end{aligned}$$

By ignoring constant terms, the second-order Taylor expansion of our surrogate term $\tilde{\Psi}_{d,b}$ is given by

$$\tilde{\Psi}_{d,b}(w_{\mathrm{L}}, w_{\mathrm{R}}) \approx \tilde{G}_{\mathrm{L}} \cdot w_{\mathrm{L}} + \frac{1}{2} \cdot \tilde{H}_{\mathrm{L}} \cdot w_{\mathrm{L}}^2 + \tilde{G}_{\mathrm{R}} \cdot w_{\mathrm{R}} + \frac{1}{2} \cdot \tilde{H}_{\mathrm{R}} \cdot w_{\mathrm{R}}^2 + \tilde{H}_{\mathrm{B}} \cdot w_{\mathrm{L}} \cdot w_{\mathrm{R}}.$$

Here, each term is defined as follows:

$$\tilde{G}_{\mathrm{L}} := \sum_{n \in \mathcal{N}_{\mathrm{L}}} \frac{\partial}{\partial w} l(+1, w) \mid_{w = \xi_t(\boldsymbol{x}_n)} + \sum_{n \in \mathcal{N}_{\mathrm{B}}} \frac{\partial}{\partial w} \tilde{l}_\beta(w, \xi_t(\boldsymbol{x}_n); \boldsymbol{x}_n) \mid_{w = \xi_t(\boldsymbol{x}_n)},$$

$$\tilde{H}_{\mathrm{L}} := \sum_{n \in \mathcal{N}_{\mathrm{L}}} \frac{\partial^2}{\partial w^2} l(+1, w) \mid_{w = \xi_t(\boldsymbol{x}_n)} + \sum_{n \in \mathcal{N}_{\mathrm{B}}} \frac{\partial^2}{\partial w^2} \tilde{l}_\beta(w, \xi_t(\boldsymbol{x}_n); \boldsymbol{x}_n) \mid_{w = \xi_t(\boldsymbol{x}_n)},$$

$$\tilde{H}_{\mathrm{B}} := \sum_{n \in \mathcal{N}_{\mathrm{B}}} \frac{\partial^2}{\partial w \partial w'} \tilde{l}_\beta(w, w'; \boldsymbol{x}_n) \mid_{w = \xi_t(\boldsymbol{x}_n), w' = \xi_t(\boldsymbol{x}_n)}.$$

We define $\tilde{G}_R$ and $\tilde{H}_R$ in a similar way. By minimizing the second-order approximation of the objective function in the inner problem of (6), we obtain an approximate solution as a closed-form:

$$w_L^* = \frac{\gamma \cdot \tilde{H}_B \cdot (G_R + \gamma \cdot \tilde{G}_R) - (H_R + \gamma \cdot \tilde{H}_R) \cdot (G_L + \gamma \cdot \tilde{G}_L)}{(H_L + \gamma \cdot \tilde{H}_L) \cdot (H_R + \gamma \cdot \tilde{H}_R) - \gamma^2 \cdot \tilde{H}_B^2}, \tag{7}$$

$$w_R^* = \frac{\gamma \cdot \tilde{H}_B \cdot (G_L + \gamma \cdot \tilde{G}_L) - (H_L + \gamma \cdot \tilde{H}_L) \cdot (G_R + \gamma \cdot \tilde{G}_R)}{(H_L + \gamma \cdot \tilde{H}_L) \cdot (H_R + \gamma \cdot \tilde{H}_R) - \gamma^2 \cdot \tilde{H}_B^2}. \tag{8}$$

As with the standard unconstrained case, a naive computation of $w_L^*$ and $w_R^*$ requires $\mathcal{O}(N)$ time due to each of the terms such as $\tilde{G}_L$ and $\tilde{H}_L$. However, we can compute them in amortized constant time by the same technique for classification trees proposed by Kanamori et al. [30]. Proposition 2 shows that we can compute an approximate solution to the problem (5) in the same time complexity as the standard gradient boosting algorithm. We present our algorithm and proof in Appendix A.

**Proposition 2.** *There exists an algorithm that approximately solves the problem* (5) *in* $\mathcal{O}(N \cdot D)$.

## 4 Recourse-aware leaf refinement

This section presents a post-processing approach for improving the recourse guarantee of a learned tree ensemble model. While minimizing the recourse loss $l_\beta$ encourages a model $f$ to ensure the existence of recourse actions for instances in a training sample $S$, there is no guarantee that the model $f$ can ensure for unseen test instances as well [51]. To alleviate this issue, we introduce a post-processing task, called *leaf refinement* [49], that modifies the leaf weights of a learned tree ensemble model $f$ under the constraint on the recourse loss $l_\beta$. Then, we show that we can guarantee the existence of recourse actions for unseen test instances through a PAC-style bound [43]. Note that our post-processing method can be applied to any tree ensemble model, and does not require any assumptions on the cost function $c$ and feasible action set $\mathcal{A}(x)$.

### 4.1 Formulation and optimization

We introduce a post-processing task that refines the leaf weights of a learned tree ensemble model $f$ under the constraint on the recourse loss $l_\beta$. Let $J = \sum_{t=1}^{T} I_t$ be the total number of leaves in the ensemble. From the definition of tree ensemble models, we can express a given learned tree ensemble model $f$ as a form of $f(x) = \sum_{j=1}^{J} w_j \cdot \phi_j(x)$ [49]. Let $\alpha \in \mathbb{R}^J$ be a vector of refined leaf weights, and we define a refined model by $f_\alpha(x) := \sum_{j=1}^{J} \alpha_j \cdot \phi_j(x)$. Given a sample $S = \{(x_n, y_n)\}_{n=1}^{N}$, we formulate our task as a constrained empirical risk minimization problem [6] defined as follows:

$$\min_{\alpha \in \mathbb{R}^J} \sum_{n=1}^{N} l(y_n, f_\alpha(x_n)) \quad \text{subject to} \quad \frac{1}{N} \sum_{n=1}^{N} l_\beta(x_n \mid f_\alpha) \leq \varepsilon, \tag{9}$$

where $\varepsilon \geq 0$ is a given parameter. Note that each leaf indicator $\phi_j$ is fixed here, and a model $f_\alpha$ can be regarded as a linear model with respect to the binary representation vector $(\phi_1(x), \ldots, \phi_J(x))$.

To make the constraint of the problem (9) tractable, we assume a deterministic oracle algorithm $A_\beta^*$ that takes an instance $x$ and returns a feasible action $a \in \mathcal{A}_\beta(x)$ based on the learned model $f$, as with the previous study [51]. Note that we can employ any existing algorithm for tree ensembles (e.g., [5, 11, 57]) and do not require the oracle to be optimal in the sense of the problem (1). We also note that we can easily extend the setting where the oracle $A_\beta^*$ returns multiple diverse actions [44].

Using the oracle algorithm $A_\beta^*$, by definition, $l_\beta(x \mid f_\alpha) \leq l(+1, f_\alpha(x + A_\beta^*(x)))$ holds for any $x$. It implies that the recourse loss $l_\beta$ for $x$ is upper bounded by the standard loss $l$ for a labeled instance $(x + A_\beta^*(x), +1)$, which is tractable to be minimized. Based on this fact, we consider the following unconstrained problem instead of (9):

$$\min_{\alpha \in \mathbb{R}^J} \sum_{n=1}^{N} l(y_n, f_\alpha(x_n)) + \lambda \cdot \sum_{n=1}^{N} l(+1, f_\alpha(x_n + A_\beta^*(x_n))), \tag{10}$$

where $\lambda \geq 0$ is a Lagrangian multiplier. There exists $\lambda$ such that the solution to the problem (10) satisfies $\frac{1}{N} \sum_{n=1}^{N} l(+1, f_\alpha(x_n + A_\beta^*(x_n))) \leq \varepsilon$, which implies that $\frac{1}{N} \sum_{n=1}^{N} l_\beta(x_n \mid f_\alpha) \leq \varepsilon$ holds

as well. Furthermore, the problem (10) can be easily solved by any off-the-shelf library since it can be regarded as a learning problem for a linear model with the weighted empirical risk on the concatenated sample $S \cup S'$, where $S' := \{(\boldsymbol{x}_n + A_\beta^*(\boldsymbol{x}_n), +1)\}_{n=1}^N$. In a nutshell, by iteratively solving the problem (10) and updating $\lambda$, we can efficiently obtain a feasible solution to the problem (9).

### 4.2 PAC-style guarantee

Our leaf refinement approach enables us to control the ratio that the refined model $f_{\boldsymbol{\alpha}}$ can ensure actions for instances in $S$ with a given parameter $\varepsilon$. However, it is not guaranteed that the model $f_{\boldsymbol{\alpha}}$ can ensure actions for unseen test instances as well. To analyze this risk, we show a PAC-style bound on the estimation error of our surrogate risk $\tilde{\mathcal{R}}_\beta(f_{\boldsymbol{\alpha}} \mid S) := \frac{1}{N} \sum_{n=1}^N l(+1, f_{\boldsymbol{\alpha}}(\boldsymbol{x}_n + A_\beta^*(\boldsymbol{x}_n)))$.

**Proposition 3.** *For a model $f \colon \mathcal{X} \to \mathbb{R}$, let $\mathcal{R}_\beta(f) := \mathbb{P}_x[\forall \boldsymbol{a} \in \mathcal{A}_\beta(\boldsymbol{x}) : \mathrm{sgn}(f(\boldsymbol{x} + \boldsymbol{a})) \neq +1]$ be the expected recourse risk of $f$. Given a sample $S = \{(\boldsymbol{x}_n, y_n)\}_{n=1}^N$, refined tree ensemble model $f_{\boldsymbol{\alpha}}(\boldsymbol{x}) = \sum_{j=1}^J \alpha_j \cdot \phi_j(\boldsymbol{x})$, and $\delta > 0$, the following inequality holds with probability at least $1 - \delta$:*

$$\mathcal{R}_\beta(f_{\boldsymbol{\alpha}}) \leq \tilde{\mathcal{R}}_\beta(f_{\boldsymbol{\alpha}} \mid S) + \sqrt{\frac{8 \cdot \ln \frac{e \cdot N}{4}}{N}} + \sqrt{\frac{\ln \frac{1}{\delta}}{2 \cdot N}}.$$

We give our proof of Proposition 3 in Appendix A. From Proposition 3, we can probably obtain a refined model $f_{\boldsymbol{\alpha}}$ that can guarantee the existence of actions for any instance with probability at least $1 - \varepsilon'$ by solving the problem (9), where $\varepsilon' = \varepsilon + \sqrt{\frac{8 \cdot \ln \frac{e \cdot N}{4}}{N}} + \sqrt{\frac{\ln \frac{1}{\delta}}{2 \cdot N}}$.

## 5 Experiments

To investigate the performance of our RABIT, we conducted experiments on real datasets. All the code was implemented in Python 3.10 with Numba 0.61.0 and is available at `https://github.com/kelicht/rabit`. All the experiments were conducted on macOS Sequoia with Apple M2 Ultra CPU and 128 GB memory. Our experimental evaluation aims to answer the following questions: (i) How are the predictive accuracy and recourse guarantee of tree ensembles learned by our RABIT compared to those by the baselines? (ii) Can our RABIT balance the trade-off between accuracy and recourse? (iii) Is our post-processing approach effective in improving the recourse guarantee of tree ensembles? Due to page limitations, the complete results are shown in Appendix C.

**Experimental settings** We used four real benchmark datasets: FICO ($N = 9871, D = 23$) [14], COMPAS ($N = 6167, D = 14$) [2], Adult ($N = 48842, D = 16$) [34], and Bail ($N = 8923, D = 16$) [53]. These datasets represent diverse real-world applications (finance, criminal justice, census, and bail prediction) that are widely used in the literature of algorithmic recourse [32]. For each dataset, all the categorical features were one-hot encoded. To obtain an action $\boldsymbol{a}$ for each instance $\boldsymbol{x}$ by solving the problem (1), we employed the *feature tweaking algorithm* [57], which is a fast heuristic algorithm for tree ensemble models. Note that we also employed the exact method based on integer optimization [11, 28], and its results are presented in Appendix C. As a cost function $c$, we used the *max percentile shift (MPS)* [58] defined as $c(\boldsymbol{a} \mid \boldsymbol{x}) = \max_{d \in [D]} |Q_d(x_d + a_d) - Q_d(x_d)|$, where $Q_d$ is the cumulative distribution function of a feature $d$. In addition, actionability constraints were imposed on certain features (e.g., gender), as shown in Appendix C. To the best of our knowledge, there is no existing method that can train GBDTs while guaranteeing the existence of actions. Following the previous studies [12, 30], we compared our RABIT with two baselines: standard unconstrained learning (*Vanilla*) and learning with only actionable features (*OAF*).

### 5.1 Baseline comparison

First, we evaluate the performance of tree ensemble models learned by our RABIT in comparison with the baselines. We randomly split the dataset into the training and test sets with a ratio of $75 : 25$, and trained tree ensemble models by each method on the training set. For learned models, we measured (1) the accuracy on the test set, (2) the recourse ratio, which is defined as the ratio of the test instances that are guaranteed valid actions whose costs are less than $\beta = 0.2$, and (3) the running time for training. We repeated this procedure 10 times. For the baselines and our RABIT, we trained $T = 100$

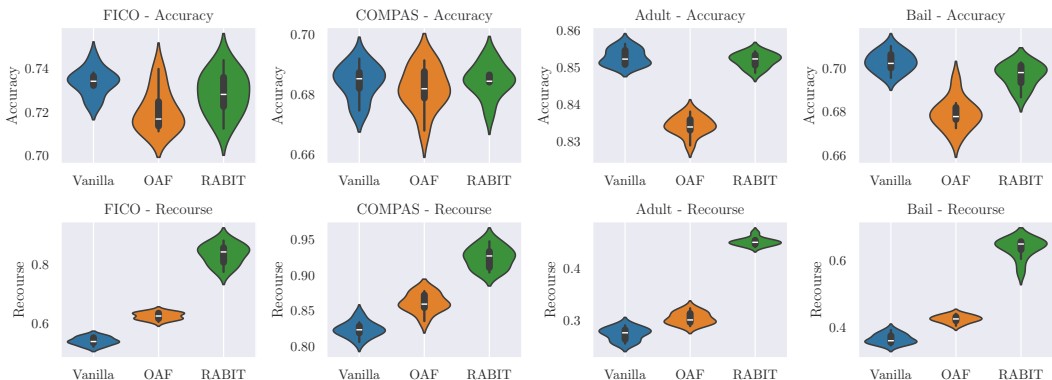

Figure 1: Experimental results of baseline comparison with respect to the accuracy and the recourse ratio (higher is better). We observed that our RABIT attained a higher recourse ratio than the baselines while keeping comparable accuracy on all the datasets.

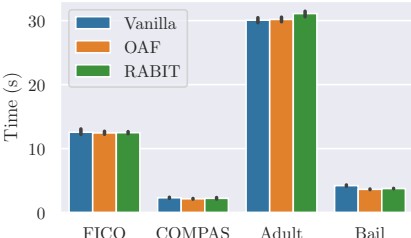

Figure 2: Average running time [s] of each method (lower is better). There is no significant difference between the baselines and our RABIT on all the datasets.

regression trees with a maximum depth of $8$ and a learning rate $0.1$. For RABIT, we set $\gamma = 0.002$ and did not apply our leaf refinement post-processing.

Figure 1 presents the results on the accuracy and recourse ratio. From Figure 1, we observed that (i) RABIT attained comparable accuracy to the baselines, and (ii) RABIT achieved significantly higher recourse ratios than the baselines. Compared to Vanilla, while OAF attained a slightly higher recourse ratio, it suffered from a significant degradation in accuracy. One possible reason is that OAF was restricted to using only actionable features, sacrificing predictive power by ignoring potentially highly predictive non-actionable features. On the other hand, our RABIT succeeded in improving the recourse guarantee without significantly degrading the predictive performance. Figure 2 shows the average running time for each dataset. We can see that there is no significant difference in the running time between the baselines and RABIT. These results indicate that our algorithm performed as fast as the standard learning algorithms, even though it additionally considers the recourse loss.

In summary, we have confirmed that *our RABIT succeeded in guaranteeing the existence of recourse actions for more instances than the baselines without compromising predictive accuracy and computational efficiency*. Our findings show that RABIT improves the availability of executable recourse actions for individuals without sacrificing the predictive accuracy or computational efficiency of standard GBDTs. This makes RABIT a practical and reliable tool for high-stakes automated decision-making systems, enhancing transparency and user trust in real-world applications.

**Recourse quality analyses** To assess the quality of actions extracted from tree ensembles trained by our RABIT, we evaluate their cost, sparsity, and plausibility. To evaluate the plausibility, the previous studies often use the outlier score $q$ of $\boldsymbol{x} + \boldsymbol{a}$ [19]. Following the previous study [45], we employed isolation forests (IF) [37] as $q$. Table 1 shows the average cost $c(\boldsymbol{a} \mid \boldsymbol{x})$, sparsity $\|\boldsymbol{a}\|_0$, and plausibility $q(\boldsymbol{x} + \boldsymbol{a})$ of the obtained valid actions $\boldsymbol{a}$ for test instances $\boldsymbol{x}$. We can see that RABIT attained lower cost and sparsity than the baselines. We also observed that there is no remarkable difference in plausibility between the baselines and RABIT. From these results, we have confirmed that *our method could provide executable recourse actions without harming their plausibility*.

Table 1: Average cost, sparsity, and plausibility of extracted actions (lower is better). Our RABIT achieved lower costs and sparsity than the baselines while keeping comparable plausibility.

| Dataset | Cost (MPS) | | | Sparsity ($\ell_0$-norm) | | | Plausibility (IF) | | |
| --- | --- | --- | --- | --- | --- | --- | --- | --- | --- |
| | Vanilla | OAF | RABIT | Vanilla | OAF | RABIT | Vanilla | OAF | RABIT |
| FICO | 0.358 | 0.315 | 0.165 | 2.155 | 2.792 | 1.078 | 0.438 | 0.44 | 0.475 |
| COMPAS | 0.23 | 0.181 | 0.112 | 1.518 | 1.553 | 1.264 | 0.436 | 0.446 | 0.466 |
| Adult | 0.354 | 0.308 | 0.275 | 1.467 | 1.547 | 1.422 | 0.477 | 0.464 | 0.479 |
| Bail | 0.449 | 0.371 | 0.225 | 1.546 | 1.563 | 1.131 | 0.508 | 0.503 | 0.519 |

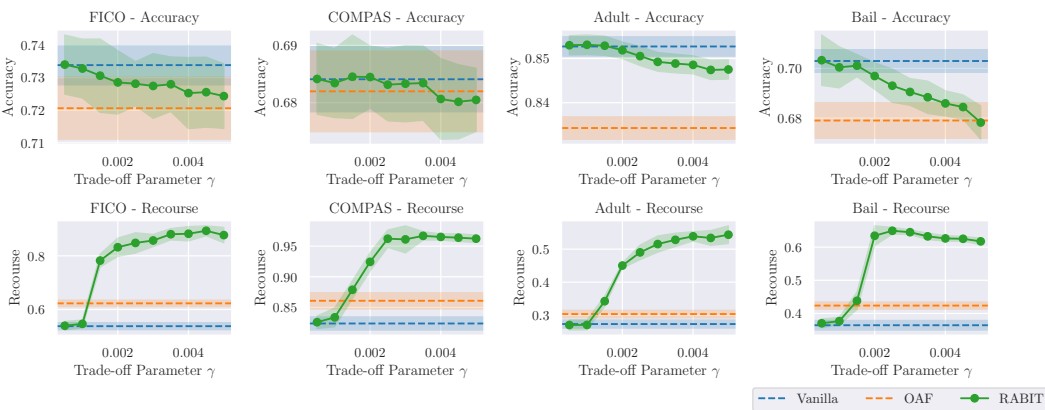

Figure 3: Sensitivity analyses of the parameter $\gamma$ with respect to the accuracy and recourse ratio. We can see that the average recourse ratio (resp. accuracy) was improved as $\gamma$ increased (resp. decreased).

## 5.2 Trade-off analysis

Next, we analyze the trade-off between the predictive accuracy and recourse guarantee of our RABIT by varying its trade-off parameter $\gamma$. Under the same experimental settings as in Section 5.1, we trained tree ensemble models by varying the trade-off parameter $\gamma$, and compared their average accuracy and recourse ratio to those of the baselines.

Figure 3 shows the average accuracy and recourse ratio for each $\gamma$. We can see that the recourse ratio (resp. accuracy) was improved by increasing (resp. decreasing) $\gamma$ on almost all the datasets, which suggests that we could balance their trade-off by tuning $\gamma$. More precisely, we observed that RABIT began to outperform the baselines in terms of the recourse ratio without significantly degrading the accuracy around $\gamma = 0.002$ on all the datasets. These results indicate that we have a chance to attain a higher recourse ratio without compromising prediction performance if we can determine the appropriate value of $\gamma$. In summary, we have confirmed that *our method could obtain models that achieve a better recourse ratio than the baselines while keeping comparable accuracy by tuning $\gamma$*.

## 5.3 Efficacy of post-processing approach

Finally, we examine the efficacy of our leaf refinement approach for improving the recourse guarantee of tree ensemble models. We randomly split the dataset into the training, calibration, and test sets with a ratio of $50 : 25 : 25$. After learning tree ensembles by each method on the training set, we applied our leaf refinement approach to them with the calibration set. Instead of solving the problem (9) directly, we solved the problem (10) for a fixed Lagrangian multiplier $\lambda$ and repeated this procedure by varying $\lambda$. We report the average accuracy and recourse ratio of the refined models over 10 trials. Note that the ablation study on our post-processing approach is presented in Appendix C.

Figure 4 presents the scatter plots of the average accuracy and recourse ratio for each $\lambda$. We can see that, in the FICO and COMPAS datasets, RABIT dominated the baselines in terms of accuracy and recourse ratio. It indicates that the combination of our learning algorithm and post-processing approach performed the best on these datasets. On the other hand, in the Adult and Bail datasets, we observed that Vanilla and RABIT demonstrated comparable performance. For example, in the

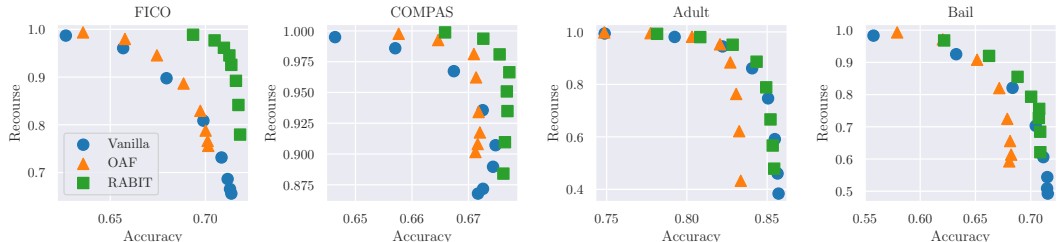

Figure 4: Experimental results of the post-processing approach. In each figure, x-axis and y-axis stand for the average accuracy and the recourse ratio, respectively. Our RABIT attained better or comparable trade-offs between the accuracy and the recourse ratio than the baselines.

Adult dataset, they maintained the accuracy around $85\%$ while improving the recourse ratio from roughly $40\%$ to $80\%$, which are better results than those in Figure 1. These results suggest that our post-processing approach was sometimes effective even for tree ensembles learned by existing standard algorithms. In summary, we have confirmed that *our post-processing approach succeeded in improving the recourse guarantee of tree ensembles while maintaining their predictive accuracy*.

## 6 Conclusion

This paper proposed a new framework of gradient boosted decision trees, named recourse-aware gradient boosted decision trees (RABIT), that can provide both accurate predictions and executable recourse actions. We proposed an efficient gradient boosting algorithm for learning tree ensemble models with the recourse loss that encourages the existence of recourse actions, and showed that its computational complexity is equivalent to that of the standard unconstrained algorithm. We also proposed a post-processing approach that refines the leaf weights of a learned tree ensemble model under the constraint on the recourse loss, and gave a PAC-style guarantee on the existence of recourse actions. Experimental results demonstrated that our RABIT succeeded in guaranteeing recourse actions for more individuals than the baselines while keeping comparable accuracy and efficiency.

**Limitations and future work**

There are several directions to make our RABIT more practical. First, the computational efficiency of our algorithm relies on our assumption of the $\ell_\infty$-type cost function $c$. For the $\ell_\infty$-type cost functions, we can easily decide whether the budget constraint is violated or not by checking each feature independently. In general, such a property does not hold for other cost functions, including $\ell_1$- or $\ell_2$-type cost functions. However, we expect that our algorithm can efficiently handle general cost functions by exploiting some heuristic strategies (e.g., changing $\beta$ depending on the tree depth [63]).

Second, deriving a tight bound for our recourse loss $l_\beta$ is important for future work. From Figure 3, we found that a slight emphasis on $l_\beta$, i.e., setting $\gamma = 0.002$, is sufficient to achieve substantial improvements in the recourse ratio without harming performance. It suggests that our bound in Proposition 1 is too conservative, and our method can be improved by developing a tighter bound.

Third, identifying when our post-processing method becomes effective is interesting for future work. In Figure 4 and our ablation study in Appendix C, we observed that the combination of our learning algorithm and post-processing method worked the best on some datasets, while it was comparable to the combination of standard learning algorithms and our post-processing method on other datasets. It suggests that there exist situations where our post-processing alone yields sufficient improvement in recourse guarantee. Such situations might be characterized by some specific properties of datasets, such as the ratio of actionable features or the percentage of categorical features. However, it is still challenging to identify a general cause of this phenomenon, and we leave it as future work.

Finally, there is room for our implementation of RABIT to be more sophisticated. In particular, our RABIT has the potential to improve scalability, generalization performance, and stability by several techniques implemented in modern frameworks of gradient boosting, such as histogram-based learning [7, 33], stochastic sampling [17, 25], and so on [35, 38]. Since these techniques are compatible with RABIT, it is worth investigating their effectiveness in our framework.

## Acknowledgments

We wish to thank Yuichi Ike for making a number of valuable suggestions. We also thank the anonymous reviewers for their insightful comments. This work was supported in part by JST ACT-X JPMJAX23C6 and JSPS KAKENHI Grant-in-Aid for Early-Career Scientists 24K17465.

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

## A  Omitted proofs

### A.1  Proof of Proposition 1

To prove Proposition 1, we first show a lemma that provides a lower bound on the maximum of the sum of two functions.

**Lemma 1.** *For any functions $g_1$ and $g_2$ defined on a finite set $X$, the following inequality holds:*

$$\max_{x \in X} (g_1(x) + g_2(x)) \geq \min_{x \in X} g_1(x) + \max_{x \in X} g_2(x).$$

*Proof.* Let $x^* = \arg\max_{x \in X} g_2(x)$. From the property of the maximum operator, we have $\max_{x \in X}(g_1(x) + g_2(x)) \geq g_1(x^*) + g_2(x^*)$. Since $g_1(x^*) \geq \min_{x \in X} g_1(x)$ and $g_2(x^*) = \max_{x \in X} g_2(x)$ hold by definition, we have $g_1(x^*) + g_2(x^*) \geq \min_{x \in X} g_1(x) + \max_{x \in X} g_2(x)$, which concludes the proof. $\square$

Using Lemma 1, we give a proof of Proposition 1 as follows.

*Proof of Proposition 1.* Recall that the loss function $l(y, \hat{y})$ is non-increasing with respect to $\hat{y}$ when $y = +1$. That is, $l(+1, \hat{y}) \leq l(+1, \hat{y}')$ holds if $\hat{y} \geq \hat{y}'$. Combining Lemma 1 with this fact, we have

$$l_\beta(\boldsymbol{x} \mid F_{t-1} + h) = \min_{\boldsymbol{a} \in \mathcal{A}_\beta(\boldsymbol{x})} l(+1, F_{t-1}(\boldsymbol{x} + \boldsymbol{a}) + h(\boldsymbol{x} + \boldsymbol{a}))$$
$$= l(+1, \max_{\boldsymbol{a} \in \mathcal{A}_\beta(\boldsymbol{x})} F_{t-1}(\boldsymbol{x} + \boldsymbol{a}) + h(\boldsymbol{x} + \boldsymbol{a}))$$
$$\leq l(+1, \min_{\boldsymbol{a} \in \mathcal{A}_\beta(\boldsymbol{x})} F_{t-1}(\boldsymbol{x} + \boldsymbol{a}) + \max_{\boldsymbol{a} \in \mathcal{A}_\beta(\boldsymbol{x})} h(\boldsymbol{x} + \boldsymbol{a}))$$

From the definition of $F_{t-1}$, we have $\min_{\boldsymbol{a} \in \mathcal{A}_\beta(\boldsymbol{x})} F_{t-1}(\boldsymbol{x} + \boldsymbol{a}) \geq \sum_{s=1}^{t-1} \min_{\boldsymbol{a} \in \mathcal{A}_\beta(\boldsymbol{x})} f_s(\boldsymbol{x} + \boldsymbol{a}) = \xi_t(\boldsymbol{x})$. Thus, we have

$$l_\beta(\boldsymbol{x} \mid F_{t-1} + h) \leq l(+1, \min_{\boldsymbol{a} \in \mathcal{A}_\beta(\boldsymbol{x})} F_{t-1}(\boldsymbol{x} + \boldsymbol{a}) + \max_{\boldsymbol{a} \in \mathcal{A}_\beta(\boldsymbol{x})} h(\boldsymbol{x} + \boldsymbol{a}))$$
$$\leq l(+1, \sum_{s=1}^{t-1} \min_{\boldsymbol{a} \in \mathcal{A}_\beta(\boldsymbol{x})} f_s(\boldsymbol{x} + \boldsymbol{a}) + \max_{\boldsymbol{a} \in \mathcal{A}_\beta(\boldsymbol{x})} h(\boldsymbol{x} + \boldsymbol{a}))$$
$$= l(+1, \xi_t(\boldsymbol{x}) + \max_{\boldsymbol{a} \in \mathcal{A}_\beta(\boldsymbol{x})} h(\boldsymbol{x} + \boldsymbol{a}))$$
$$= \min_{\boldsymbol{a} \in \mathcal{A}_\beta(\boldsymbol{x})} l(+1, \xi_t(\boldsymbol{x}) + h(\boldsymbol{x} + \boldsymbol{a})),$$

which concludes the proof. $\square$

### A.2  Proof of Proposition 2

**Algorithm**  We present our algorithm for approximately solving the problem (5) in Algorithm 1. In addition to the subset of a sample $\mathcal{N}(r_{t,i})$ and trade-off parameter $\gamma$, our algorithm takes the following inputs:

1. a set of candidate thresholds $B_d = \{b_{d,1}, \ldots, b_{d,M_d}\}$ such that $b_{d,1} \leq \cdots \leq b_{d,M_d}$ and $|B_d| = \mathcal{O}(N)$ for each feature $d \in [D]$;

2. a permutation $\sigma_d$ such that $x_{\sigma_d(1),d} \leq \cdots \leq x_{\sigma_d(N),d}$ for each $d \in [D]$;

3. gradient and hessian statistics for each $n \in \mathcal{N}(r_{t,i})$ defined as follows:

$$g_n := \frac{\partial}{\partial \hat{y}} l(y_n, \hat{y}) \mid_{\hat{y} = F_{t-1}(\boldsymbol{x}_n)}, \quad h_n := \frac{\partial^2}{\partial \hat{y}^2} l(y_n, \hat{y}) \mid_{\hat{y} = F_{t-1}(\boldsymbol{x}_n)},$$

$$\hat{g}_n := \frac{\partial}{\partial \hat{y}} l(+1, \hat{y}) \mid_{\hat{y} = \xi_t(\boldsymbol{x}_n)}, \quad \hat{h}_n := \frac{\partial^2}{\partial \hat{y}^2} l(+1, \hat{y}) \mid_{\hat{y} = \xi_t(\boldsymbol{x}_n)},$$

$$\bar{g}_n := \frac{\partial}{\partial w} \tilde{l}_\beta(w, \xi_t(\boldsymbol{x}_n); \boldsymbol{x}_n) \mid_{w = \xi_t(\boldsymbol{x}_n)} \quad \bar{h}_n := \frac{\partial^2}{\partial w^2} \tilde{l}_\beta(w, \xi_t(\boldsymbol{x}_n); \boldsymbol{x}_n) \mid_{w = \xi_t(\boldsymbol{x}_n)},$$

$$\tilde{h}_n := \frac{\partial^2}{\partial w \partial w'} \tilde{l}_\beta(w, w'; \boldsymbol{x}_n) \mid_{w = \xi_t(\boldsymbol{x}_n), w' = \xi_t(\boldsymbol{x}_n)} . \tag{11}$$

4. indicator values $o_n(d, b)$ and $\bar{o}_n(d, b)$ for each $n \in \mathcal{N}(r_{t,i})$, $d \in [D]$, and $b \in B_d$, which are defined as follows:

$$o_n(d, b) := \mathbb{I}\left[\min_{\boldsymbol{a} \in \mathcal{A}_\beta(\boldsymbol{x}_n)} a_d \leq b - x_{n,d}\right], \quad \bar{o}_n(d, b) := \mathbb{I}\left[\max_{\boldsymbol{a} \in \mathcal{A}_\beta(\boldsymbol{x}_n)} a_d > b - x_{n,d}\right]. \tag{12}$$

**Algorithm 1** Algorithm for approximately solving the problem (5).

---

1: $G \leftarrow \sum_{n \in \mathcal{N}(r_{t,i})} g_n$; $H \leftarrow \sum_{n \in \mathcal{N}(r_{t,i})} h_n$; $\tilde{G} \leftarrow \sum_{n \in \mathcal{N}(r_{t,i})} \hat{g}_n$; $\tilde{H} \leftarrow \sum_{n \in \mathcal{N}(r_{t,i})} \hat{h}_n$;
2: **for** $d = 1, 2, \ldots, D$ **do**
3:   /* Initialize each term */
4:   $G_{\mathrm{L}}, H_{\mathrm{L}}, \tilde{G}_{\mathrm{L}}, \tilde{H}_{\mathrm{L}}, \tilde{H}_{\mathrm{B}} \leftarrow 0, 0, 0, 0, 0$; $G_{\mathrm{R}}, H_{\mathrm{R}}, \tilde{G}_{\mathrm{R}}, \tilde{H}_{\mathrm{R}} \leftarrow G, H, \tilde{G}, \tilde{H}$;
5:   $m, m_{\mathrm{L}}, m_{\mathrm{R}} \leftarrow 1$; $n, n_{\mathrm{L}}, n_{\mathrm{R}} \leftarrow \sigma_d(m), \sigma_d(m_{\mathrm{L}}), \sigma_d(m_{\mathrm{R}})$;
6:   **for** $b \in B_d$ **do**
7:     /* Update terms corresponding to the standard loss $l$ */
8:     **while** $x_{n,d} \leq b$ and $m \leq N$ **do**
9:       $G_{\mathrm{L}}, G_{\mathrm{R}}, H_{\mathrm{L}}, H_{\mathrm{R}} \leftarrow G_{\mathrm{L}} + g_n, G_{\mathrm{R}} - g_n, H_{\mathrm{L}} + h_n, H_{\mathrm{R}} - h_n$;
10:       $m \leftarrow m + 1$; $n \leftarrow \sigma_d(m)$;
11:     **end while**
12:     /* Update terms corresponding to the recourse loss $l_\beta$ */
13:     **while** $o_{n_{\mathrm{L}}}(d, b) = 1$ and $m_{\mathrm{L}} \leq N$ **do**
14:       $\tilde{G}_{\mathrm{L}}, \tilde{H}_{\mathrm{L}} \leftarrow \tilde{G}_{\mathrm{L}} + \hat{g}_{n_{\mathrm{L}}} - \bar{g}_{n_{\mathrm{L}}}, \tilde{H}_{\mathrm{L}} + \hat{h}_{n_{\mathrm{L}}} - \bar{h}_{n_{\mathrm{L}}}$;
15:       $\tilde{H}_{\mathrm{B}} \leftarrow \tilde{H}_{\mathrm{B}} - \tilde{h}_{n_{\mathrm{L}}}$;
16:       $m_{\mathrm{L}} \leftarrow m_{\mathrm{L}} + 1$; $n_{\mathrm{L}} \leftarrow \sigma_d(m_{\mathrm{L}})$;
17:     **end while**
18:     **while** $\bar{o}_{n_{\mathrm{R}}}(d, b) = 0$ and $m_{\mathrm{R}} \leq N$ **do**
19:       $\tilde{G}_{\mathrm{R}}, \tilde{H}_{\mathrm{R}} \leftarrow \tilde{G}_{\mathrm{R}} - \hat{g}_{n_{\mathrm{R}}} + \bar{g}_{n_{\mathrm{R}}}, \tilde{H}_{\mathrm{R}} - \hat{h}_{n_{\mathrm{R}}} + \bar{h}_{n_{\mathrm{R}}}$;
20:       $\tilde{H}_{\mathrm{B}} \leftarrow \tilde{H}_{\mathrm{B}} + \tilde{h}_{n_{\mathrm{R}}}$;
21:       $m_{\mathrm{R}} \leftarrow m_{\mathrm{R}} + 1$; $n_{\mathrm{R}} \leftarrow \sigma_d(m_{\mathrm{R}})$;
22:     **end while**
23:     /* Compute leaf weights and approximate loss */
24:     Compute leaf weights $w_{\mathrm{L}}(d, b)$ and $w_{\mathrm{R}}(d, b)$ by (7);
25:     Compute approximate objective value $\mathrm{loss}(d, b, w_{\mathrm{L}}(d, b), w_{\mathrm{R}}(d, b))$ by (13);
26:   **end for**
27: **end for**
28: /* Find the best parameters */
29: $d^*, b^* \leftarrow \arg\min_{d \in [D], b \in B_d} \mathrm{loss}(d, b, w_{\mathrm{L}}(d, b), w_{\mathrm{R}}(d, b))$;
30: $w_{\mathrm{L}}^*, w_{\mathrm{R}}^* \leftarrow w_{\mathrm{L}}(d^*, b^*), w_{\mathrm{R}}(d^*, b^*)$;
31: **return** $d^*, b^*, w_{\mathrm{L}}^*, w_{\mathrm{R}}^*$;

---

Note that while we need to compute these inputs in advance, it roughly takes at most $\mathcal{O}(D \cdot N^2)$ and we only need to compute them once as pre-processing before growing trees [30].

In the following, we give a proof of Proposition 2.

*Proof of Proposition 2.* To prove Proposition 2, we first prove that Algorithm 1 computes an approximate solution to the problem (5), and then, we show that it runs in $\mathcal{O}(D \cdot N)$ time.

**Correctness**  We first show that Algorithm 1 returns a split condition $(d, b)$ and leaf weights $w_{\mathrm{L}}, w_{\mathrm{R}}$ that minimize our approximate objective function of the problem (5) by the same technique proposed by [30]. By applying the second-order Taylor expansion to our surrogate objective function in (6), we obtain our approximate objective function as follows:

$$
\begin{aligned}
\mathrm{loss}(d, b, w_{\mathrm{L}}, w_{\mathrm{R}}) =& (G_{\mathrm{L}} + \gamma \cdot \tilde{G}_{\mathrm{L}}) \cdot w_{\mathrm{L}} + \frac{1}{2} \cdot (H_{\mathrm{L}} + \gamma \cdot \tilde{H}_{\mathrm{L}}) \cdot w_{\mathrm{L}}^2 \\
&+ (G_{\mathrm{R}} + \gamma \cdot \tilde{G}_{\mathrm{R}}) \cdot w_{\mathrm{R}} + \frac{1}{2} \cdot (H_{\mathrm{R}} + \gamma \cdot \tilde{H}_{\mathrm{R}}) \cdot w_{\mathrm{R}}^2 + \gamma \cdot \tilde{H}_{\mathrm{B}} \cdot w_{\mathrm{L}} \cdot w_{\mathrm{R}}.
\end{aligned}
\tag{13}
$$

For a fixed split condition $(d, b)$, we can compute the optimal leaf weights that minimize (13) by taking the derivatives of (13) with respect to $w_{\mathrm{L}}$ and $w_{\mathrm{R}}$ and setting it to zero, which yields our closed-form solution shown in (7) of the main paper. Since the candidate features $d \in [D]$ and thresholds $b \in B_d$ are finite, we can obtain the optimal split condition $(d, b)$ and leaf weights $w_{\mathrm{L}}, w_{\mathrm{R}}$ that minimize (13) by enumerating all the possible split conditions, computing their optimal leaf weights, and comparing their objective values $\mathrm{loss}(d, b, w_{\mathrm{L}}, w_{\mathrm{R}})$. Thus, in the following, we prove

that Algorithm 1 exactly computes each term in (13) for each split condition $(d, b)$. For that purpose, we show that we can compute the terms $\tilde{G}_\mathrm{L}, \tilde{H}_\mathrm{L}, \tilde{H}_\mathrm{B}$ for a split condition $(d, b_m)$ using those for the previous split condition $(d, b_{m-1})$, which we denote $\tilde{G}'_\mathrm{L}, \tilde{H}'_\mathrm{L}, \tilde{H}'_\mathrm{B}$. For notational simplicity, we assume $\mathcal{N}(r_{t,i}) = [N]$ and $x_{1,d} < \cdots < x_{N,d}$ without loss of generality. Recall that $v_\beta(\boldsymbol{x}_n; r_{t,i}) = 1$ holds for any $n \in [N]$, and thus, we have $v_\mathrm{L}(\boldsymbol{x}_n) = v_\beta(\boldsymbol{x}_n; r_{t,i}) \cdot o_n(d, b_m) = o_n(d, b_m)$ and $v_\mathrm{R}(\boldsymbol{x}_n) = v_\beta(\boldsymbol{x}_n; r_{t,i}) \cdot \bar{o}_n(d, b_m) = \bar{o}_n(d, b_m)$ by definition. In addition, we can see the two monotonic properties on $n \in [N]$ and $m \in [M_d]$: $o_n(d, b_m) \geq o_{n'}(d, b_m)$ for any $n' > n$ and $o_n(d, b_m) \geq o_n(d, b_{m'})$ for any $m' < m$. While the former implies that $o_n(d, b_m) = 0 \implies o_{n'}(d, b_m) = 0$ holds for any $n' > n$, the latter implies that $o_n(d, b_m) = 1 \implies o_n(d, b_{m'}) = 1$ holds for any $m' > m$. Let $k = \min_{n \in [N]: o_n(d, b_{m-1}) = 0} n$ and $k' = \max_{n \in [N]: o_n(d, b_m) = 1} n$. Combining the above properties and the definitions of $\tilde{G}_\mathrm{L}, \tilde{H}_\mathrm{L}, \tilde{H}_\mathrm{B}$, we have

$$\tilde{G}_\mathrm{L} - \tilde{G}'_\mathrm{L} = \sum_{n=k}^{k'} \hat{g}_n - \sum_{n=k}^{k'} \bar{g}_n \iff \tilde{G}_\mathrm{L} = \tilde{G}'_\mathrm{L} + \sum_{n=k}^{k'} (\hat{g}_n - \bar{g}_n),$$

$$\tilde{H}_\mathrm{L} - \tilde{H}'_\mathrm{L} = \sum_{n=k}^{k'} \hat{h}_n - \sum_{n=k}^{k'} \bar{h}_n \iff \tilde{H}_\mathrm{L} = \tilde{H}'_\mathrm{L} + \sum_{n=k}^{k'} (\hat{h}_n - \bar{h}_n),$$

$$\tilde{H}_\mathrm{B} - \tilde{H}'_\mathrm{B} = -\sum_{n=k}^{k'} \tilde{h}_n \iff \tilde{H}_\mathrm{B} = \tilde{H}'_\mathrm{B} - \sum_{n=k}^{k'} \tilde{h}_n.$$

Because the similar monotonic properties hold for $\bar{o}_n$, we can compute $\tilde{G}_\mathrm{R}$ and $\tilde{H}_\mathrm{R}$ in the same way. Note that $G_\mathrm{L}, H_\mathrm{L}, G_\mathrm{R}, H_\mathrm{R}$ can also be computed in a similar manner, as shown in [7]. Algorithm 1 updates each term of (13) in lines 8–22 using these facts. To conclude, we can see that Algorithm 1 correctly computes the leaf weights for each split condition, and thus, it returns the optimal split condition $(d, b)$ and leaf weights $w_\mathrm{L}, w_\mathrm{R}$ that minimize our surrogate objective function.

**Complexity**  We now show that Algorithm 1 runs in $\mathcal{O}(D \cdot N)$ time. Our complexity analysis of Algorithm 1 can be divided into the following four parts:

- In line 1, Algorithm 1 initializes the terms in $\mathcal{O}(N)$.

- In the for-loop of lines 6–26, each while-loop runs at most $N$ times through the for-loop of a fixed $B_d$. In addition, we can compute leaf weights $w_\mathrm{L}, w_\mathrm{R}$ and their objective value in constant time. Since we assume $|B_d| = \mathcal{O}(N)$, the overall complexity is $\mathcal{O}(N)$.

- Since the inner for-loop of lines 6–26 takes $\mathcal{O}(N)$, the outer for-loop of lines 2–27 takes $\mathcal{O}(D \cdot N)$.

- The optimization task in line 29 can be solved in $\mathcal{O}(D \cdot N)$ because the objective value of each $(d, b)$ has been computed in lines 2–27.

In summary, the overall complexity of Algorithm 1 is $\mathcal{O}(D \cdot N)$, which concludes the proof. $\qquad \square$

### A.3  Proof of Proposition 3

To prove Proposition 3, we first show two lemmas. Our first lemma is a Rademacher complexity bound for the expected recourse risk $\mathcal{R}_\beta(f)$.

**Lemma 2.** *For a model class $\mathcal{F}$, let $\Omega(\mathcal{F})$ be the Rademacher complexity of $\mathcal{F}$. Then, for any $f \in \mathcal{F}$ and $\delta > 0$, the following inequality holds with probability at least $1 - \delta$:*

$$\mathcal{R}_\beta(f) \leq \hat{\mathcal{R}}_\beta(f \mid S) + \Omega(\mathcal{F}) + \sqrt{\frac{\ln \frac{1}{\delta}}{2 \cdot N}},$$

*where $\hat{\mathcal{R}}_\beta(f \mid S) := \frac{1}{N} \sum_{n=1}^{N} l_{01}(+1, f(\boldsymbol{x} + A^*_\beta(\boldsymbol{x})))$.*

*Proof.* Let $\mathcal{D}$ be a distribution over the input domain $\mathcal{X}$. By the definition of the expected recourse risk $\mathcal{R}_\beta$, we have

$$\mathcal{R}_\beta(f) = \mathbb{P}_{\boldsymbol{x} \sim \mathcal{D}} [\forall \boldsymbol{a} \in \mathcal{A}_\beta(\boldsymbol{x}) : \mathrm{sgn}(f(\boldsymbol{x} + \boldsymbol{a})) \neq +1]$$
$$= \mathbb{E}_{\boldsymbol{x} \sim \mathcal{D}} [\mathbb{I} [\forall \boldsymbol{a} \in \mathcal{A}_\beta(\boldsymbol{x}) : \mathrm{sgn}(f(\boldsymbol{x} + \boldsymbol{a})) \neq +1]]$$
$$= \mathbb{E}_{\boldsymbol{x} \sim \mathcal{D}} [\min_{\boldsymbol{a} \in \mathcal{A}_\beta(\boldsymbol{x})} l_{01}(+1, f(\boldsymbol{x} + \boldsymbol{a}))].$$

From our assumptions on the oracle $A_\beta^*$, we have $\min_{a \in \mathcal{A}_\beta(x)} l_{01}(+1, f(x + a)) \le l_{01}(+1, f(x + A_\beta^*(x)))$ for any $x \in \mathcal{X}$. Thus, we have $\mathcal{R}_\beta(f) \le \mathbb{E}_{x \sim \mathcal{D}}[l_{01}(+1, f(x + A_\beta^*(x)))]$. By applying the Rademacher complexity bound [43], we have

$$\mathbb{E}_{x \sim \mathcal{D}}[l_{01}(+1, f(x + A_\beta^*(x)))] \le \frac{1}{N} \sum_{n=1}^{N} l_{01}(+1, f(x + A_\beta^*(x))) + \Omega(\mathcal{F}) + \sqrt{\frac{\ln \frac{1}{\delta}}{2 \cdot N}}$$

with probability at least $1 - \delta$. Combining these inequalities, the following inequality holds with probability at least $1 - \delta$:

$$\mathcal{R}_\beta(f) \le \mathbb{E}_{x \sim \mathcal{D}}[l_{01}(+1, f(x + A_\beta^*(x)))]$$

$$\le \frac{1}{N} \sum_{n=1}^{N} l_{01}(+1, f(x + A_\beta^*(x))) + \Omega(\mathcal{F}) + \sqrt{\frac{\ln \frac{1}{\delta}}{2 \cdot N}} = \hat{\mathcal{R}}_\beta(f \mid S) + \Omega(\mathcal{F}) + \sqrt{\frac{\ln \frac{1}{\delta}}{2 \cdot N}},$$

which concludes the proof. $\qquad\square$

In our second lemma, we show that the Rademacher complexity of the refined model $f_\alpha$ is equivalent to that of the leaf indicator functions $\phi_j$ if we normalize the weight vector $\alpha$.

**Lemma 3.** *Let $\mathcal{H} := \{\phi_1, \ldots, \phi_J\}$ and $\bar{\mathcal{F}} := \{f_\alpha \mid \|\alpha\|_1 \le 1\}$, where $f_\alpha(x) = \sum_{j=1}^{J} \alpha_j \cdot \phi_j(x)$. Then, we have $\Omega(\bar{\mathcal{F}}) = \Omega(\mathcal{H})$.*

*Proof.* Recall that the Rademacher complexity $\Omega(\mathcal{F})$ of a model class $\mathcal{F}$ is defined as $\Omega(\mathcal{F}) := \mathbb{E}_{S,\sigma}[\sup_{f \in \mathcal{F}} \frac{1}{N} \sum_{n=1}^{N} \sigma_n \cdot f(x_n)]$ [43]. Thus, we have

$$\Omega(\bar{\mathcal{F}}) = \mathbb{E}_{S,\sigma} \left[ \sup_{f_\alpha \in \bar{\mathcal{F}}} \frac{1}{N} \sum_{n=1}^{N} \sigma_n \cdot f_\alpha(x_n) \right]$$

$$= \mathbb{E}_{S,\sigma} \left[ \sup_{\alpha \in \mathbb{R}^J : \|\alpha\|_1 \le 1} \frac{1}{N} \sum_{n=1}^{N} \sigma_n \cdot \sum_{j=1}^{J} \alpha_j \cdot \phi_j(x_n) \right]$$

$$= \mathbb{E}_{S,\sigma} \left[ \sup_{\alpha \in \mathbb{R}^J : \|\alpha\|_1 \le 1} \frac{1}{N} \sum_{j=1}^{J} \alpha_j \cdot \sum_{n=1}^{N} \sigma_n \cdot \phi_j(x_n) \right]$$

$$= \mathbb{E}_{S,\sigma} \left[ \max_{j \in [J]} \frac{1}{N} \sum_{n=1}^{N} \sigma_n \cdot \phi_j(x_n) \right]$$

$$= \mathbb{E}_{S,\sigma} \left[ \sup_{\phi \in \mathcal{H}} \frac{1}{N} \sum_{n=1}^{N} \sigma_n \cdot \phi(x_n) \right] = \Omega(\mathcal{H}),$$

where the fourth equality holds by the property of the dual norm [43]. $\qquad\square$

Using Lemmas 2 and 3, we give a proof of Proposition 3 as follows.

*Proof of Proposition 3.* For any $\alpha \in \mathbb{R}^J$, let $\bar{\alpha} := \frac{1}{\|\alpha\|_1} \alpha$. Since $\|\bar{\alpha}\|_1 = 1$, $f_{\bar{\alpha}} \in \bar{\mathcal{F}}$ holds. Thus, from Lemmas 2 and 3, we have

$$\mathcal{R}_\beta(f_{\bar{\alpha}}) \le \hat{\mathcal{R}}_\beta(f_{\bar{\alpha}} \mid S) + \Omega(\mathcal{H}) + \sqrt{\frac{\ln \frac{1}{\delta}}{2 \cdot N}}$$

with probability at least $1 - \delta$. Recall that $\phi_j \in \mathcal{H}$ is defined as $\phi_j(x) = \mathbb{I}[x \in r_j]$ with an axis-aligned rectangle $r_j \subseteq \mathcal{X}$. Hence, the VC dimension of $\mathcal{H}$ is 4 and its Rademacher complexity is bounded by $\Omega(\mathcal{H}) \le \sqrt{\frac{8 \cdot \ln \frac{e \cdot N}{4}}{N}}$ [43]. In addition, since $f_{\bar{\alpha}}(x) = \frac{1}{\|\alpha\|_1} f_\alpha(x)$ holds, we have $\text{sgn}(f_{\bar{\alpha}}(x)) = \text{sgn}(f_\alpha(x))$, which implies $l_{01}(+1, f_{\bar{\alpha}}(x + A_\beta^*(x))) = l_{01}(+1, f_\alpha(x + A_\beta^*(x)))$. Thus, $\mathcal{R}_\beta(f_{\bar{\alpha}}) = \mathcal{R}_\beta(f_\alpha)$ and $\hat{\mathcal{R}}_\beta(f_{\bar{\alpha}} \mid S) = \hat{\mathcal{R}}_\beta(f_\alpha \mid S)$ holds. Finally, recall that $l_{01}(+1, f(x + A_\beta^*(x))) \le l(+1, f(x + A_\beta^*(x)))$ holds from our assumption on the loss function $l$. Therefore, we

---

**Algorithm 2** Actionable feature tweaking algorithm for approximately solving the problem (1).

1: $\boldsymbol{a}^* \leftarrow 0$; $c^* \leftarrow \infty$;
2: /* Enumerate all leaves in ensemble */
3: **for** $t = 1, \ldots, T$ **do**
4:     **for** $i = 1, \ldots, I_t$ **do**
5:         /* Optimize action for leaf */
6:         $\hat{\boldsymbol{a}} \leftarrow \arg\min_{\boldsymbol{a} \in \mathcal{A}(\boldsymbol{x})} c(\boldsymbol{a} \mid \boldsymbol{x})$ s.t. $\boldsymbol{x} + \boldsymbol{a} \in r_{t,i}$;
7:         /* Check if the action is better */
8:         **if** $\text{sgn}(f(\boldsymbol{x} + \hat{\boldsymbol{a}})) = +1$ and $c(\hat{\boldsymbol{a}} \mid x) < c^*$ **then**
9:             $\boldsymbol{a}^* \leftarrow \hat{\boldsymbol{a}}$; $c^* \leftarrow c(\hat{\boldsymbol{a}} \mid \boldsymbol{x})$;
10:         **end if**
11:     **end for**
12: **end for**
13: **return** $\boldsymbol{a}^*$;

---

have $\hat{\mathcal{R}}_\beta(f_{\bar{\boldsymbol{\alpha}}} \mid S) \leq \tilde{\mathcal{R}}_\beta(f_{\bar{\boldsymbol{\alpha}}} \mid S)$ by definition. To summarize, the following inequality holds with probability at least $1 - \delta$:

$$\mathcal{R}_\beta(f_{\boldsymbol{\alpha}}) = \mathcal{R}_\beta(f_{\bar{\boldsymbol{\alpha}}})$$

$$\leq \hat{\mathcal{R}}_\beta(f_{\bar{\boldsymbol{\alpha}}} \mid S) + \Omega(\mathcal{H}) + \sqrt{\frac{\ln\frac{1}{\delta}}{2 \cdot N}}$$

$$\leq \hat{\mathcal{R}}_\beta(f_{\bar{\boldsymbol{\alpha}}} \mid S) + \sqrt{\frac{8 \cdot \ln\frac{e \cdot N}{4}}{N}} + \sqrt{\frac{\ln\frac{1}{\delta}}{2 \cdot N}}$$

$$= \hat{\mathcal{R}}_\beta(f_{\boldsymbol{\alpha}} \mid S) + \sqrt{\frac{8 \cdot \ln\frac{e \cdot N}{4}}{N}} + \sqrt{\frac{\ln\frac{1}{\delta}}{2 \cdot N}}$$

$$\leq \tilde{\mathcal{R}}_\beta(f_{\boldsymbol{\alpha}} \mid S) + \sqrt{\frac{8 \cdot \ln\frac{e \cdot N}{4}}{N}} + \sqrt{\frac{\ln\frac{1}{\delta}}{2 \cdot N}},$$

which concludes the proof. $\qquad\square$

## B  Implementation Details

### B.1  Baseline methods

**Vanilla.** To the best of our knowledge, there is no study on learning tree ensemble models by gradient boosting while guaranteeing the existence of recourse actions. Thus, as baseline approaches, we employed standard unconstrained gradient boosting methods, which we refer to as *Vanilla*. Note that it can be implemented by simply setting $\gamma = 0.0$ in our RABIT.

**Only actionable features (OAF).** As another baseline, we employed a modified version of Vanilla, named *only actionable features (OAF)*, that uses only features that can be changed by actions. Note that such a baseline was employed by the previous studies as well [12, 30]. Its idea is based on the observation of the existing study that relying on actionable features facilitates the existence of actions [12]. Our setting of actionability of the features in each dataset is shown in Tables 2 to 5.

### B.2  Actionable feature tweaking algorithm

To extract actions from tree ensemble models $f(\boldsymbol{x}) = \sum_{t=1}^{T} f_t(\boldsymbol{x})$ by solving the problem (1), we employed *actionable feature tweaking algorithm* [57] which is a fast heuristic method. Algorithm 2 presents a pseudo-code of the actionable feature tweaking algorithm. Algorithm 2 consists of the following three steps: (i) for each tree $f_t$ in the ensemble, enumerating all the leaves $i \in [I_t]$; (ii) computing an optimal action $\hat{\boldsymbol{a}}$ to the region $r_{t,i}$ corresponding to the leaf $i$; (iii) finding the minimum cost action $\boldsymbol{a}^*$ among ones altering the prediction results of $f$ into the desired class (i.e.,

$\mathrm{sgn}(f(\boldsymbol{x} + \boldsymbol{a})) = +1)$. Note that we can easily compute an optimal action $\hat{\boldsymbol{a}} = (\hat{a}_1, \ldots, \hat{a}_D)$ to an instance $\boldsymbol{x}$ and a region $r = [l_1, u_1] \times \cdots \times [l_D, u_D]$ as $\hat{a}_d = \mathrm{median}(x_d, l_d, u_d) - x_d$ for $d \in [D]$ [5, 63]. Our implementation is parallelized and runs faster than the existing public ones.

## C    Complete experimental results

We present the complete results of our experiments. Tables 2 to 5 present the details on the value type, minimum value, maximum value, immutability, and constraint of each feature of the datasets that we used in our experiments.

### C.1    Additional experimental results of baseline comparison

Figure 5 presents the experimental results of our baseline comparison with respect to the area under the ROC curve (AUC) and F1 score. As with the results with respect to the accuracy, we observed that RABIT achieved comparable AUC and F1 score to the baselines. Table 6 shows the average cost, sparsity, and plausibility of extracted actions with their standard deviations.

### C.2    Sensitivity analysis of hyperparameters

**Number of trees $T$**    Figure 6 presents the experimental results of our baseline comparison with different numbers of trees $T$. We varied $T$ from 50 to 250 with a step size of 50, and measured the average accuracy and recourse ratio for each $T$. We can see that RABIT attained higher recourse ratios than the baselines while keeping comparable accuracy regardless of the number of trees $T$.

**Cost budget $\beta$**    Figure 7 presents the sensitivity analysis of the cost budget $\beta$. We varied $\beta$ from 0.1 to 0.5 with a step size of 0.1, and measured the average accuracy and recourse ratio for each $\beta$. We can see that RABIT stably attained comparable accuracy to the baselines regardless of the cost budget $\beta$. We also observed that RABIT achieved higher recourse ratios than the baselines for almost all $\beta$.

### C.3    Efficacy of intercept adjustment

By transforming a learned tree ensemble $f$ into a probabilistic forecaster $g_\theta(\boldsymbol{x}) = \frac{1}{1+e^{-f(\boldsymbol{x})+\theta}}$ with an intercept $\theta \in \mathbb{R}$, we can directly apply the existing post-processing method that adjusts the intercept $\theta$ through a PAC-style guarantee [51]. To investigate the efficacy of the existing intercept adjustment, we applied it to the baselines and RABIT and measured the accuracy and recourse ratio of the adjusted models.

Figure 8 presents the results, where we fixed $\delta = 0.05$ and varying $\varepsilon$ from 0.05 to 0.5 with a step size of 0.05. We can see that while the recourse ratios of all the methods could be controlled by the parameter $\varepsilon$, RABIT succeeded in maintaining the best accuracy for almost all the situations. However, compared to our leaf refinement approach, the intercept adjustment approach tended to drop the accuracy significantly when $\varepsilon$ is small. For the example on the Adult dataset, the accuracies of the intercept adjustment approach were less than 50% when $\varepsilon \leq 0.1$, while those of our leaf refinement approach were more than 80% as shown in Figure 4. These results suggest that the intercept adjustment approach is effective in controlling the recourse ratio, but it is not as effective as our leaf refinement approach in terms of accuracy.

### C.4    Impact on model brittleness

We examine the impact of our method on the model brittleness, i.e., the sensitivity of the model to small perturbations of the input data. Following the previous work [51], we add i.i.d. Gaussian noises to test instances and measure the ratio of instances whose predictions change. For each test instance, we generate 100 perturbed instances by adding $\delta_d \sim \mathcal{N}(0, 0.1 \cdot \sigma_d^2)$ for each actionable feature $d$, where $\sigma_d^2$ is the variance of $d$.

Table 7 presents the results of the average model brittleness of each method. We can see that there is no significant difference in the model brittleness between RABIT and the baselines, which suggests that our method does not significantly affect the model brittleness.

### C.5 Comparison using Exact AR Algorithm based on Integer Optimization

To investigate whether an action extraction algorithm affects the results, we also employed the exact AR method based on integer optimization [11, 28] to extract actions from tree ensembles trained by each method. As with our experiment of Section 5.1, we extracted actions for test instances predicted as the undesired class in each trial, and measured the average cost of the extracted actions. We used PySCIPOpt 5.5.0, which is a Python wrapper of SCIP[1], one of the fastest open-source mathematical programming solvers. Due to computational cost, we randomly picked 30 test instances in each fold, and a 60 second time limit was imposed on optimizing an action for each instance.

Table 8 presents the average cost of the extracted actions. As with our results using the heuristic algorithm shown in Section 5.1, we can see that our RABIT achieved lower cost than the baselines regardless of the datasets.

### C.6 Ablation Study on Leaf Refinement

To investigate the efficacy of our learning algorithm and leaf refinement approach, we conducted its ablation study that compares Vanilla and RABIT with Vanilla with leaf refinement (*Vanilla w/ Refinement*) and RABIT with leaf refinement (*RABIT w/ Refinement*). Note that there are two main evaluation metrics (i.e., accuracy and recourse ratio) and the parameter $\lambda$ of our post-hoc refinement affects these metrics. To make the comparison easier to understand, for each method and dataset, we selected the value of $\lambda$ that attained the closest accuracy to that of RABIT.

Table 9 presents the average accuracy and recourse ratio of each method. We can see that the accuracy gaps between methods were at most 2.7%, 0.6%, 0.3%, and 1.0% on FICO, COMPAS, Adult, and Bail datasets, respectively. It indicates that *all the methods attained comparable accuracy in each dataset*. In addition, we observed that RABIT w/ Refinement performed the best, and that RABIT and Vanilla w/ Refinement performed better than Vanilla. Furthermore, we also observed that while RABIT outperformed Vanilla w/ Refinement in FICO and COMPAS datasets, Vanilla w/ Refinement outperformed RABIT in Adult and Bail datasets. These results indicate that our learning algorithm and post-hoc refinement are effectice individually, and their combination can achieve better results.

## D  Additional Comments on Existing Assets

Numba 0.61.0[2] is publicly available under the BSD-2-Clause license. PySCIPOpt 5.5.0[3] is publicly available under the MIT license. All the scripts and datasets used in our experiments are available in our GitHub repository at `https://github.com/kelicht/rabit`.

All the datasets used in Section 5 are publicly available and do not contain any identifiable information or offensive content. As they are accompanied by appropriate citations in the main body, see the corresponding references for more details.

## E  Discussion on potential societal impacts

Our proposed method, named RABIT, is a new framework that aims to learn accurate tree ensemble models while guaranteeing the existence of recourse actions. As demonstrated in our experiments, tree ensembles trained by our method can provide executable actions to more individuals than the existing baselines without degrading accuracy. Thus, our method enables us to learn predictive models that make accurate predictions and guarantee recourse actions. It improves the trustworthiness of algorithmic decision-making for critical tasks in the real world, such as loan approvals and judicial decisions [36, 58].

On the other hand, our framework also has potential societal impacts that need careful consideration. In practice, it may not always be necessary for decision-makers to guarantee the existence of executable actions for all individuals. For example, providing easy actions for granting loans to applicants who do not have sufficient capacity to repay might cause a serious financial crisis in the

---

[1]`https://www.scipopt.org/`
[2]`https://numba.pydata.org/`
[3]`https://pyscipopt.readthedocs.io/`

future. However, since our method can adjust the ratio of individuals who are guaranteed to have executable actions by tuning its hyperparameter $\gamma$, it helps decision-makers provide recourse actions to appropriate individuals while keeping the quality of decision-making.

In addition, there is a risk that our method may be used maliciously to train a model that provides specific actions for causing some undesired situations, such as discrimination. To avoid this risk, we may need to check the actions provided to affected individuals before deploying the models (e.g., using the existing techniques for globally summarizing recourse actions [29, 48]).

Overall, the proposed method has the potential to significantly improve the trustworthiness of the decision-making process, but we need careful consideration of its risks before incorporating it into the actual decision-making process.

Table 2: Details of each feature of the FICO dataset [14].

| Feature | Type | Min | Max | Immutable | Constraint |
|---|---|---|---|---|---|
| ExternalRiskEstimate | Integer | 0.00 | 94.00 | Yes | Fix |
| MSinceOldestTradeOpen | Integer | 0.00 | 803.00 | Yes | Fix |
| MSinceMostRecentTradeOpen | Integer | 0.00 | 383.00 | Yes | Fix |
| AverageMInFile | Integer | 4.00 | 383.00 | Yes | Fix |
| NumSatisfactoryTrades | Integer | 0.00 | 79.00 | No | Nothing |
| NumTrades60Ever2DerogPubRec | Integer | 0.00 | 19.00 | Yes | Fix |
| NumTrades90Ever2DerogPubRec | Integer | 0.00 | 19.00 | Yes | Fix |
| PercentTradesNeverDelq | Integer | 0.00 | 100.00 | No | Nothing |
| MSinceMostRecentDelq | Integer | 0.00 | 83.00 | No | Nothing |
| MaxDelq2PublicRecLast12M | Integer | 0.00 | 9.00 | No | Nothing |
| MaxDelqEver | Integer | 2.00 | 8.00 | No | Nothing |
| NumTotalTrades | Integer | 0.00 | 104.00 | Yes | Fix |
| NumTradesOpeninLast12M | Integer | 0.00 | 19.00 | Yes | Fix |
| PercentInstallTrades | Integer | 0.00 | 100.00 | No | Nothing |
| MSinceMostRecentInqexcl7days | Integer | 0.00 | 24.00 | No | Nothing |
| NumInqLast6M | Integer | 0.00 | 66.00 | No | Nothing |
| NumInqLast6Mexcl7days | Integer | 0.00 | 66.00 | No | Nothing |
| NetFractionRevolvingBurden | Integer | 0.00 | 232.00 | No | Nothing |
| NetFractionInstallBurden | Integer | 0.00 | 471.00 | No | Nothing |
| NumRevolvingTradesWBalance | Integer | 0.00 | 32.00 | No | Nothing |
| NumInstallTradesWBalance | Integer | 0.00 | 23.00 | No | Nothing |
| NumBank2NatlTradesWHighUtilization | Integer | 0.00 | 18.00 | No | Nothing |
| PercentTradesWBalance | Integer | 0.00 | 100.00 | No | Nothing |

Table 3: Details of each feature of the COMPAS dataset [2].

| Feature | Type | Min | Max | Immutable | Constraint |
|---|---|---|---|---|---|
| age | Integer | 18.00 | 96.00 | No | Irreducible |
| juv_fel_count | Integer | 0.00 | 20.00 | No | Nothing |
| juv_misd_count | Integer | 0.00 | 13.00 | No | Nothing |
| juv_other_count | Integer | 0.00 | 17.00 | No | Nothing |
| priors_count | Integer | 0.00 | 38.00 | No | Nothing |
| sex:Female | Binary | 0.00 | 1.00 | Yes | Fix |
| sex:Male | Binary | 0.00 | 1.00 | Yes | Fix |
| race:African-American | Binary | 0.00 | 1.00 | Yes | Fix |
| race:Asian | Binary | 0.00 | 1.00 | Yes | Fix |
| race:Caucasian | Binary | 0.00 | 1.00 | Yes | Fix |
| race:Hispanic | Binary | 0.00 | 1.00 | Yes | Fix |
| race:Native-American | Binary | 0.00 | 1.00 | Yes | Fix |
| race:Other | Binary | 0.00 | 1.00 | Yes | Fix |
| c_charge_degree:F | Binary | 0.00 | 1.00 | No | Nothing |
| c_charge_degree:M | Binary | 0.00 | 1.00 | No | Nothing |

Table 4: Details of each feature of the Adult dataset [34].

| Feature | Type | Min | Max | Immutable | Constraint |
|---|---|---|---|---|---|
| age | Integer | 17.00 | 90.00 | No | Irreducible |
| educational-num | Integer | 1.00 | 16.00 | No | Nothing |
| capital-gain | Integer | 0.00 | 99999.00 | No | Nothing |
| capital-loss | Integer | 0.00 | 4356.00 | No | Nothing |
| hours-per-week | Integer | 1.00 | 99.00 | No | Nothing |
| marital-status:Married | Binary | 0.00 | 1.00 | Yes | Fix |
| marital-status:NotMarried | Binary | 0.00 | 1.00 | Yes | Fix |
| race:Amer-Indian-Eskimo | Binary | 0.00 | 1.00 | Yes | Fix |
| race:Asian-Pac-Islander | Binary | 0.00 | 1.00 | Yes | Fix |
| race:Black | Binary | 0.00 | 1.00 | Yes | Fix |
| race:Other | Binary | 0.00 | 1.00 | Yes | Fix |
| race:White | Binary | 0.00 | 1.00 | Yes | Fix |
| gender:Female | Binary | 0.00 | 1.00 | Yes | Fix |
| gender:Male | Binary | 0.00 | 1.00 | Yes | Fix |
| native-country:Others | Binary | 0.00 | 1.00 | Yes | Fix |
| native-country:US | Binary | 0.00 | 1.00 | Yes | Fix |

Table 5: Details of each feature of the Bail dataset [53].

| Feature | Type | Min | Max | Immutable | Constraint |
|---|---|---|---|---|---|
| White | Binary | 0.00 | 1.00 | Yes | Fix |
| Alchy | Binary | 0.00 | 1.00 | No | Nothing |
| Junky | Binary | 0.00 | 1.00 | No | Nothing |
| Super | Binary | 0.00 | 1.00 | Yes | Fix |
| Married | Binary | 0.00 | 1.00 | Yes | Fix |
| Felon | Binary | 0.00 | 1.00 | Yes | Fix |
| Workrel | Binary | 0.00 | 1.00 | No | Nothing |
| Propty | Binary | 0.00 | 1.00 | Yes | Fix |
| Person | Binary | 0.00 | 1.00 | Yes | Fix |
| Male | Binary | 0.00 | 1.00 | Yes | Fix |
| Priors | Integer | 0.00 | 40.00 | No | Nothing |
| School | Integer | 1.00 | 19.00 | No | Nothing |
| Rule | Integer | 0.00 | 39.00 | No | Nothing |
| Age | Integer | 15.00 | 72.00 | No | Irreducible |
| Tservd | Integer | 1.00 | 287.00 | Yes | Fix |
| Follow | Integer | 46.00 | 57.00 | Yes | Fix |

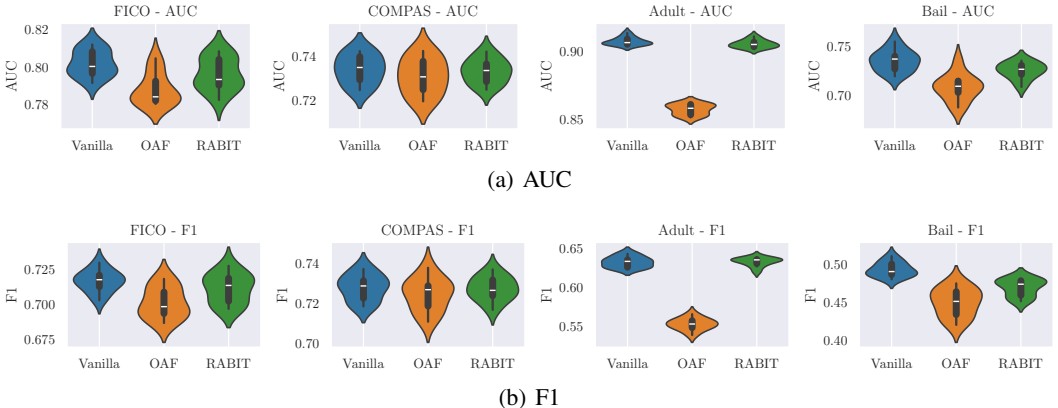

(a) AUC

(b) F1

Figure 5: Experimental results of baseline comparison with respect to the area under the ROC curve (AUC) and F1 score of the models trained by each method (higher is better).

Table 6: Average cost, sparsity, and plausibility of extracted actions with their standard deviation (lower is better).

(a) Cost

| Dataset | Vanilla | OAF | RABIT |
|---|---|---|---|
| FICO | $0.358 \pm 0.01$ | $0.315 \pm 0.01$ | $0.165 \pm 0.03$ |
| COMPAS | $0.23 \pm 0.02$ | $0.181 \pm 0.01$ | $0.112 \pm 0.01$ |
| Adult | $0.354 \pm 0.0$ | $0.308 \pm 0.01$ | $0.275 \pm 0.0$ |
| Bail | $0.449 \pm 0.02$ | $0.371 \pm 0.01$ | $0.225 \pm 0.03$ |

(b) Sparsity

| Dataset | Vanilla | OAF | RABIT |
|---|---|---|---|
| FICO | $2.155 \pm 0.06$ | $2.792 \pm 0.08$ | $1.078 \pm 0.05$ |
| COMPAS | $1.518 \pm 0.05$ | $1.553 \pm 0.05$ | $1.264 \pm 0.04$ |
| Adult | $1.467 \pm 0.07$ | $1.547 \pm 0.1$ | $1.422 \pm 0.14$ |
| Bail | $1.546 \pm 0.1$ | $1.563 \pm 0.05$ | $1.131 \pm 0.03$ |

(c) Plausibility

| Dataset | Vanilla | OAF | RABIT |
|---|---|---|---|
| FICO | $0.438 \pm 0.0$ | $0.44 \pm 0.0$ | $0.475 \pm 0.0$ |
| COMPAS | $0.436 \pm 0.01$ | $0.446 \pm 0.01$ | $0.466 \pm 0.01$ |
| Adult | $0.477 \pm 0.01$ | $0.464 \pm 0.01$ | $0.479 \pm 0.01$ |
| Bail | $0.508 \pm 0.0$ | $0.503 \pm 0.0$ | $0.519 \pm 0.0$ |

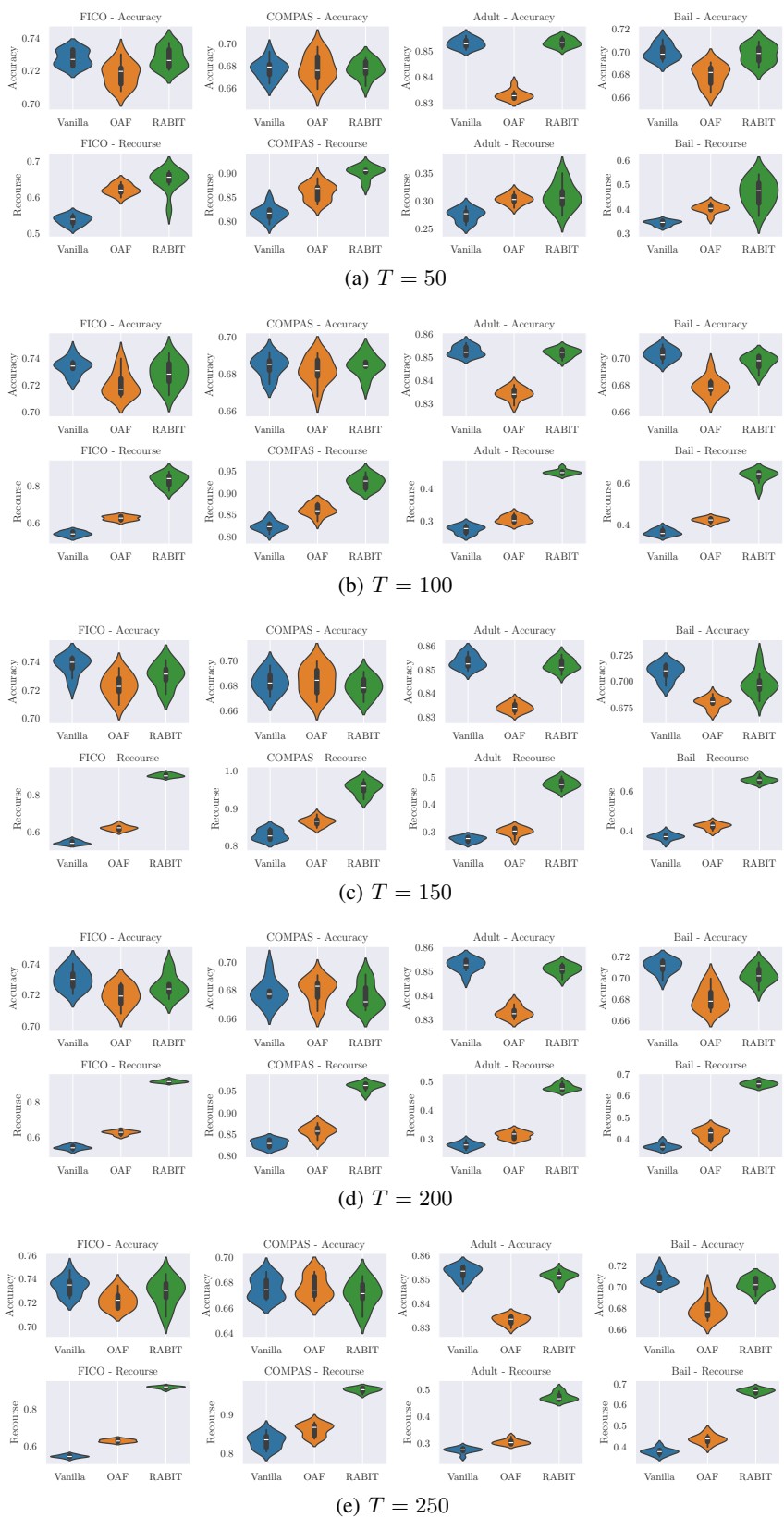

Figure 6: Sensitivity analysis of the number of trees $T$.

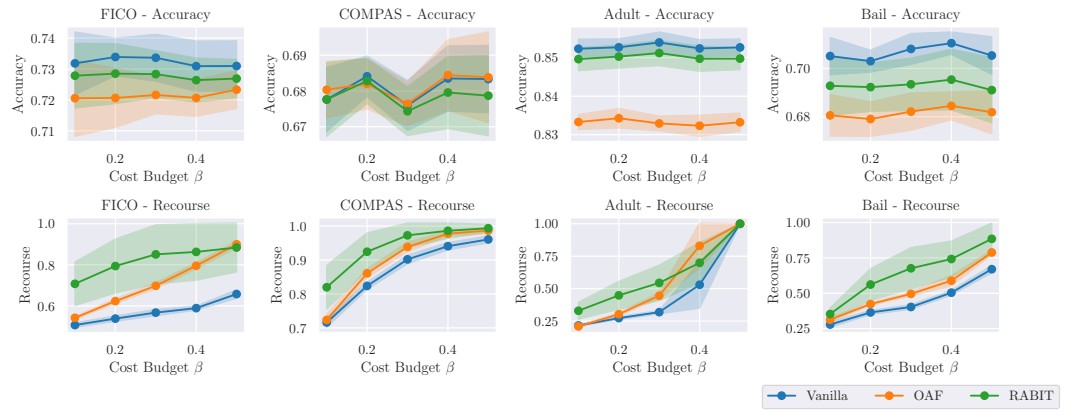

Figure 7: Sensitivity analysis of the cost budget $\beta$.

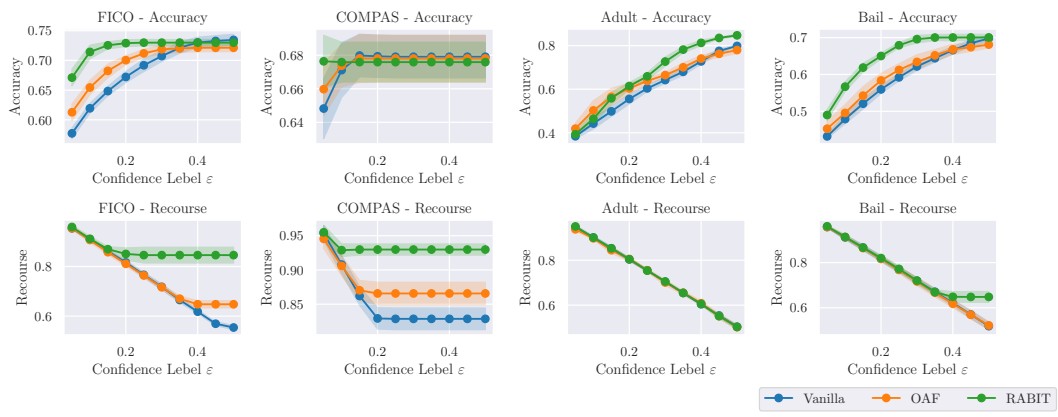

Figure 8: Experimental results of the intercept adjustment approach proposed by [51].

Table 7: Average brittleness of the models trained by each method against input perturbations (lower is better).

| Dataset | Vanilla | OAF | RABIT |
|---|---|---|---|
| FICO | $0.064 \pm 0.0$ | $0.098 \pm 0.01$ | $0.063 \pm 0.0$ |
| COMPAS | $0.107 \pm 0.0$ | $0.114 \pm 0.0$ | $0.113 \pm 0.0$ |
| Adult | $0.04 \pm 0.0$ | $0.064 \pm 0.0$ | $0.043 \pm 0.0$ |
| Bail | $0.112 \pm 0.01$ | $0.144 \pm 0.01$ | $0.105 \pm 0.01$ |

Table 8: Average cost of actions extracted by the exaxt AR method (lower is better).

| Dataset | Vanilla | OAF | RABIT |
|---|---|---|---|
| FICO | $0.354 \pm 0.05$ | $0.293 \pm 0.07$ | $\mathbf{0.11 \pm 0.01}$ |
| COMPAS | $0.188 \pm 0.04$ | $0.158 \pm 0.03$ | $\mathbf{0.096 \pm 0.01}$ |
| Adult | $0.34 \pm 0.02$ | $0.297 \pm 0.02$ | $\mathbf{0.257 \pm 0.02}$ |
| Bail | $0.42 \pm 0.05$ | $0.32 \pm 0.04$ | $\mathbf{0.213 \pm 0.02}$ |

Table 9: Average accuracy and recourse ratio of each method in the ablation study on our learning algorithm and leaf refinement method (higher is better).

(a) Accuracy

| Dataset | Vanilla | RABIT | Vanilla w/ Refinement | RABIT w/ Refinement |
|---------|---------|-------|----------------------|---------------------|
| FICO | $0.735 \pm 0.01$ | $0.732 \pm 0.01$ | $0.708 \pm 0.01$ | $0.716 \pm 0.01$ |
| COMPAS | $0.682 \pm 0.01$ | $0.677 \pm 0.01$ | $0.676 \pm 0.01$ | $0.677 \pm 0.01$ |
| Adult | $0.852 \pm 0.0$ | $0.851 \pm 0.0$ | $0.853 \pm 0.0$ | $0.85 \pm 0.0$ |
| Bail | $0.711 \pm 0.01$ | $0.701 \pm 0.0$ | $0.702 \pm 0.01$ | $0.706 \pm 0.01$ |

(b) Recourse Ratio

| Dataset | Vanilla | RABIT | Vanilla w/ Refinement | RABIT w/ Refinement |
|---------|---------|-------|----------------------|---------------------|
| FICO | $0.541 \pm 0.01$ | $0.825 \pm 0.03$ | $0.694 \pm 0.01$ | $0.859 \pm 0.01$ |
| COMPAS | $0.832 \pm 0.02$ | $0.936 \pm 0.02$ | $0.887 \pm 0.01$ | $0.986 \pm 0.0$ |
| Adult | $0.274 \pm 0.01$ | $0.427 \pm 0.02$ | $0.685 \pm 0.06$ | $0.847 \pm 0.05$ |
| Bail | $0.364 \pm 0.02$ | $0.628 \pm 0.03$ | $0.84 \pm 0.03$ | $0.868 \pm 0.01$ |

