# OpenReview forum: "Learning Gradient Boosted Decision Trees with Algorithmic Recourse"
_NeurIPS.cc/2025/Conference — NeurIPS 2025 poster_

### Official Review · Reviewer_nKF5 · 2025-06-26

**Clarity:** 3
**Significance:** 3
**Originality:** 3
**Rating:** 4
**Confidence:** 4

**Summary:**

This paper focuses on Algorithmic Recourse. The paper proposes a new gradient boosted decision tree framework, called Resource-Aware Gradient Boosted Decision Tree (RABIT), which can provide both accurate predictions and executable resource actions. A post-processing method is also proposed to refine the leaf weights, and a PAC-style guarantee is provided. Experimental results on four datasets show that RABIT outperforms the baseline while maintaining comparable accuracy and efficiency.

**Questions:**

1. There are few data sets used in the experiment. Why did you choose these four data sets?

2. Is it only binary classification? How about multiple classifications?

3. In the introduction section, it was mentioned that some features, such as age and race, cannot be modified. How did you reflect this in your experiment?

4. “Figure 3 shows the average accuracy and recourse ratio for each γ. We can see that the recourse ratio 294 (resp. accuracy) was improved by increasing (resp. decreasing) γ on almost all the datasets, which 295 suggests that we could balance their trade-off by tuning “ I disagree with this statement. Figure 3 can only show that there is a trade-off, but it cannot show that RABIT can balance this trade-off.

5. Are the experimental results in Sections 5.1 and 5.2 using post-processing or not? What are the ablation experiments with and without post-processing?

6. Why is the accuracy lower and the variance larger on adult and bail from t=100 to t=150? The same thing happens on FICO and COMPAS from t=150 to t=200.

**Ethical Concerns:**

["NO or VERY MINOR ethics concerns only"]

**Final Justification:**

My concerns are addressed, so I keep my positive score.

**Limitations:**

yes

**Quality:**

3

**Strengths And Weaknesses:**

## Strengths

The issues this paper focuses on are important, especially since some of the features cannot be modified as mentioned in the introduction.

The paper gives a PAC-style guarantee.

## Weaknesses

The experiments are not sufficient, for example, only four datasets are used, and the analysis of some experimental results is not very clear.

---

> ### Author Rebuttal · Authors · 2025-07-31
>
> We would like to thank the reviewer for valuable and thoughtful feedback. We will reflect all of them in our final version. In the following, we will respond to the key questions raised by the reviewer.
>
> ---
> > 1. There are few data sets used in the experiment. Why did you choose these four data sets?
>
> We chose the four datasets (FICO, COMPAS, Adult, and Bail) because they are widely used and well-established benchmarks in the literature on algorithmic recourse (e.g., Ross et al. 2021, Kanamori et al. 2024). These datasets represent diverse real-world applications (finance, criminal justice, census, and bail prediction), allowing us to demonstrate the generalizability and effectiveness of RABIT across different domains. While we acknowledge that more datasets could further strengthen our experiments, we believe these four datasets are sufficient to validate our claims and compare our method with existing baselines, as evidenced by prior studies in this field that also use these datasets.
>
>
> > 2. Is it only binary classification? How about multiple classifications?
>
> Thank you for your important question. While the current manuscript focuses on binary classification, our method can be applied to multiclass classification settings if we can reduce the multiclass problem to a binary one. For example, if we can divide the classes into desirable and undesirable ones, then the multiclass problem is reduced to a binary classification problem between these two classes.
>
>
> > 3. In the introduction section, it was mentioned that some features, such as age and race, cannot be modified. How did you reflect this in your experiment?
>
> In our experiments, we defined immutable features for each dataset, which are shown in Tables 2-5 in Appendix C. When we extracted recourse actions from the models trained by each method, we imposed the actionability constraints so that the generated actions do not change immutable features, such as age and race. In addition, our RABIT incorporates such actionability constraints in our recourse loss, which encourages the model to guarantee the existence of recourse actions that do not change immutable features. In summary, we take into account the actionability constraints during both the recourse extraction and model training phases.
>
>
> > 4. “Figure 3 shows the average accuracy and recourse ratio for each γ. We can see that the recourse ratio 294 (resp. accuracy) was improved by increasing (resp. decreasing) γ on almost all the datasets, which 295 suggests that we could balance their trade-off by tuning “ I disagree with this statement. Figure 3 can only show that there is a trade-off, but it cannot show that RABIT can balance this trade-off.
>
> We appreciate your careful reading and acknowledge the nuance in the statement. Let us clarify our claim "we could balance their trade-off" in line 295.
>
> As you pointed out, it is true that Figure 3 shows a trade-off between the recourse ratio and accuracy. However, Figure 3 also indicates the potential that we can attain a higher recourse ratio without compromising prediction performance if we can determine the appropriate value of $\gamma$. From this observation, we claim that if we execute RABIT with multiple $\gamma$, we have a chance to achieve a high recourse ratio with comparable accuracy to baselines, which is mentioned in lines 295-299. In our final version, we will rephrase these statements to be more precise and avoid any ambiguity.
>
>
> > 5. Are the experimental results in Sections 5.1 and 5.2 using post-processing or not? What are the ablation experiments with and without post-processing?
>
> Thank you for your important question. In our experiments in Sections 5.1 and 5.2, we did not use our post-processing method.
>
> While our paper does not explicitly include the ablation experiments on our post-hoc refinement, we can discuss its effect in an ablation study manner by reorganizing the results in Figures 1 and 4, where we compare the performances of Vanilla, RABIT, Vanilla w/ Refinement, and RABIT w/ Refinement.
>
> The reorganized results are shown as our reply to Reviewer bYie. As we replied to Reviewer bYie, these results indicate that our learning algorithm and post-hoc refinement are effective individually, and their combination can achieve significantly better results than the baseline. In our final version, we will add this ablation result and discuss it.
>
>
> > 6. Why is the accuracy lower and the variance larger on adult and bail from t=100 to t=150? The same thing happens on FICO and COMPAS from t=150 to t=200.
>
> Thank you for your careful reading. We think that there are some reasons for such results.
>
> One possible reason is that the trained models might have overfitted the training datasets in some cases as the total number of $T$, i.e., model complexity, increased. In our experiments, we fixed the learning rate to $0.1$ and maximum depth of trees to $8$ regardless of $T$. If we carefully tune these parameters, we may obtain more stable results.
>
> Another reason is the heuristic nature of gradient boosting algorithms. While our method provides theoretical guarantees, the practical implementation of gradient boosting involves some heuristic techniques, such as greedy splitting, approximation based on Taylor expansion, and bootstrap sampling for training each tree. These heuristics might lead to less stable solutions and increase variance.
>
> In practical situations, we acknowledge that it is valuable to stabilize our algorithm without harming its performance. In our final version, we will mention this point as future work.
>
>
> ---
> We hope that we have adequately addressed all your questions and concerns. Please let us know if we can provide any further details and/or clarifications. Thank you again for your valuable feedback.

---

> > ### Comment · Reviewer_nKF5 · 2025-08-04
> >
> > Thanks for your reply, I will keep my positive score.

---

> > > ### Author Response · Authors · 2025-08-09
> > >
> > > Thank you for your message and for keeping your positive score. In our final version, we will reflect all of your insightful suggestions and constructive feedback. We appreciate your time and valuable feedback.

---

### Official Review · Reviewer_JBHL · 2025-06-26

**Clarity:** 3
**Significance:** 2
**Originality:** 3
**Rating:** 5
**Confidence:** 3

**Summary:**

The authors propose a recourse-aware gradient-boosting algorithm (RABIT) that considers recourse actions. They also propose a post-processing task of modifying a learned tree ensemble model to satisfy the constraint on recourse guarantee and theoretically and empirically demonstrate the efficacy of RABIT in comparison to vanilla gradient boosting algorithms.

**Questions:**

- Actionability and mutability of features.
The authors consider two cases: RABIT, which places less emphasis on actionability, and OAF, which restricts itself to actionable features.
While I understand the premise of the author's argument and appreciate the two versions, it is unclear to me whether the recourse generated based on the RABIT-trained trees respects actionability constraints.
For example, even if RABIT does not explicitly incorporate actionability, do the generated recourses avoid suggesting unrealistic or non-actionable changes, such as increasing age by 10 years? While technically viable, the recommendation is not actionable and could encourage gaming behavior rather than genuine improvement.
Additionally, regarding the example on lines 31–36, would it be possible for the authors to use another example?

- Price of recourse.
The authors use the maximum percentile shift costs function. However, Ustun et al. mention that this might not correctly reflect the difficulty
of recourse. How easy(hard) would it be to equation 5 to other forms of the price of recourse?

- Experiments.
While the authors do a great job of illustrating the experimental results, the explanations of the observations are sparse.
For example, can the authors provide explanations for why OAF (Figure) consistently has the lowest accuracy and why the recourse ratio is close to 1 and 0.5 in other datasets, or why RABIT consistently has the highest ratio? Why is the gamma parameter better at 0.002 to 0.004?

**Ethical Concerns:**

["NO or VERY MINOR ethics concerns only"]

**Final Justification:**

I will retain the positive score with the hope that authors incorporate the promised changes such as a comprehensive limitations and  and future works sections, and the new observations e.g., extension of leaf refinement problem to the case where instance has multiple actions, among others.

**Limitations:**

Yes

**Paper Formatting Concerns:**

No formatting issues.

**Quality:**

3

**Strengths And Weaknesses:**

**Strengths**
- The authors clearly state the problem and present theoretically and empirically sound results.
- They theoretically prove that with and without recourse guarantees, they can get an approximate solution in O(ND) time.
- To ensure recourse guarantees for unseen test instances, authors also propose a leaf refinement approach and show a PAC-style bound on the estimation error of the surrogate risk.
- Authors empirically test RABIT on four commonly used real-world datasets, provide key details about the experimentation process in the main paper and appendix, and avail their code.

**Weaknesses**
- Actionability and mutability of features.
The authors consider two cases: RABIT, which places less emphasis on actionability, and OAF, which restricts itself to actionable features.
While I understand the premise of the author's argument and appreciate the two versions, it is unclear to me whether the recourse generated based on the RABIT-trained trees respects actionability constraints.
For example, even if RABIT does not explicitly incorporate actionability, do the generated recourses avoid suggesting unrealistic or non-actionable changes, such as increasing age by 10 years? While technically viable, the recommendation is not actionable and could encourage gaming behavior rather than genuine improvement.
Additionally, regarding the example on lines 31–36, would it be possible for the authors to use another example?

- Price of recourse.
The authors use the maximum percentile shift costs function. However, Ustun et al. mention that this might not correctly reflect the difficulty
of recourse. How easy(hard) would it be to equation 5 to other forms of the price of recourse?

- Experiments.
While the authors do a great job of illustrating the experimental results, the explanations of the observations are sparse.
For example, can the authors provide explanations for why OAF (Figure) consistently has the lowest accuracy and why the recourse ratio is close to 1 and 0.5 in other datasets, or why RABIT consistently has the highest ratio? Why is the gamma parameter better at 0.002 to 0.004?

---

> ### Author Rebuttal · Authors · 2025-07-31
>
> We would like to thank the reviewer for valuable and thoughtful feedback. We will reflect all of them in our final version. In the following, we will respond to the key comments and questions raised by the reviewer.
>
> ---
> > Actionability and mutability of features. The authors consider two cases: RABIT, which places less emphasis on actionability, and OAF, which restricts itself to actionable features. While I understand the premise of the author's argument and appreciate the two versions, it is unclear to me whether the recourse generated based on the RABIT-trained trees respects actionability constraints. For example, even if RABIT does not explicitly incorporate actionability, do the generated recourses avoid suggesting unrealistic or non-actionable changes, such as increasing age by 10 years? While technically viable, the recommendation is not actionable and could encourage gaming behavior rather than genuine improvement.
>
> Our method is designed to avoid suggesting unrealistic or non-actionable changes. This is because we explicitly incorporate actionability during the model training phase, as well as the recourse generating phase.
>
> At the training phase, we consider the learning objective function that consists of the standard loss and our recourse loss. Our recourse loss is defined as $l_\beta(x \mid f) = \min_{a \in \mathcal{A}_\beta(x)} l(+1, f(x+a))$, where the set of feasible actions $\mathcal{A}_\beta(x)$ is constructed under given actionability constraints. For example, we can enforce $\mathcal{A}_\beta(x)$ to not include actions that change age or gender. Our recourse loss encourages learning a model whose decision boundary is "recourse-friendly" within the predefined actionability constraints, rather than simply ignoring immutable features like OAF. By minimizing our learning objective function, we can guarantee the existence of realistic and actionable recourse actions while maintaining sufficient accuracy by exploiting both actionable and immutable features.
>
> In addition, at the generating phase, we extract actions from the trained models under the actionability constraints so that the generated actions do not change immutable features. Note that this can be done by existing algorithms for generating recourse actions. Our main contribution is to incorporate actionability in the training phase of gradient boosted decision trees and demonstrate that we can ensure the existence of realistic actions for many instances without degrading predictive accuracy and computational efficiency.
>
>
> > Additionally, regarding the example on lines 31–36, would it be possible for the authors to use another example?
>
> We can provide other examples with respect to lines 31-36. For example, "past bankruptcy" is an important feature for estimating the risk of default, but it might be immutable for a certain period. Another example is "number of dependents," which might be practically difficult to change solely for the purpose of a loan. In our final version, we will revise our example scenario.
>
>
> > Price of recourse. The authors use the maximum percentile shift costs function. However, Ustun et al. mention that this might not correctly reflect the difficulty of recourse. How easy(hard) would it be to equation 5 to other forms of the price of recourse?
>
> Thank you for your important comment. The computational efficiency of our algorithm relies on a property of $\ell_\infty$-type cost functions, including the maximum percentile shift. Thus, it would be challenging to maintain its efficiency by simply extending our method to other cost functions.
>
> Specifically, for $\ell_\infty$-type cost functions, we can easily decide whether the budget constraint is violated or not by checking each feature independently. In general, this property does not hold for other cost functions such as $\ell_1$-norm. Thus, it is not trivial to deal with these cost functions in our current approach while maintaining its computational efficiency.
>
> In practice, however, we expect that if we introduce some heuristic strategies such as changing the budget parameter $\beta$ for each depth of tree (Wang et al. 2020), our approach can be extended to deal with other cost functions without harming efficiency. In our final version, we will clarify this limitation of our current algorithm and mention it as future work.
>
>
> > Experiments. While the authors do a great job of illustrating the experimental results, the explanations of the observations are sparse. For example, can the authors provide explanations for why OAF (Figure) consistently has the lowest accuracy and why the recourse ratio is close to 1 and 0.5 in other datasets, or why RABIT consistently has the highest ratio? Why is the gamma parameter better at 0.002 to 0.004?
>
> We appreciate your important comment on our explanations of the observations. In our final version, we will add more detailed discussions on our experimental results.
>
> **On the lowest accuracy of OAF**:
> One possible reason why OAF consistently had the lowest accuracy is that OAF restricts models to only actionable features during training, sacrificing predictive power by ignoring potentially highly predictive non-actionable features. For example, in the Adult dataset, OAF could use only 5 of the 16 features and could not use immutable features such as gender, race, and age, which might degrade the accuracy of OAF compared to other methods. We think this inherent limitation might lead to lower accuracy compared to models using all features.
>
> **On the Variance of Recourse Ratio and Highest Ratio of RABIT**:
> The recourse ratio depends heavily on the inherent properties of the datasets. In particular, we guess that it might depend on the specific actionability constraints defined for each feature (as shown in Tables 2-5 in Appendix C). For instance, in FICO, many financial features (e.g., NumSatisfactoryTrades, NetFractionRevolvingBurden) are actionable, making it easier to find low-cost recourses for almost all instances. In contrast, since Adult includes many demographic features that are not actionable, there might be no actionable change if the trained model heavily relies on these immutable features for its predictions. It suggests that it is challenging to provide actionable and low-cost actions even for half of the instances in such a dataset. As mentioned above, the objective function of our RABIT directly optimizes for recourse existence via the recourse loss term that explicitly guides the model to find recourse-friendly decision boundaries. We believe it is a main reason why our RABIT consistently had the highest recourse ratio.
>
> **On the Gamma Parameter**:
> By our definition, $\gamma$ balances predictive accuracy and recourse guarantee in our learning objective function. We observed that $\gamma = 0.002$ is a sweet spot where RABIT achieves significant improvements in recourse ratio without significantly degrading accuracy, regardless of the datasets in our experiments. It suggests that such a slight emphasis on our recourse loss is sufficient to achieve substantial improvements without harming predictive performance. One possible reason is that our bound for the recourse loss shown in Proposition 1 is too conservative. It suggests that our method can be improved by developing a tighter bound, which is interesting for future work.
>
>
> ---
> We hope that we have adequately addressed all your questions and concerns. Please let us know if we can provide any further details and/or clarifications. Thank you again for your valuable feedback.

---

> > ### Comment · Reviewer_JBHL · 2025-08-04
> >
> > Thank you for addressing my questions. I will retain the positive score. I hope the authors will incorporate the identified limitations and suggested future work, as well as other promised changes or new observations, into the final version of the paper.

---

> > > ### Author Response · Authors · 2025-08-09
> > >
> > > We are very grateful for your positive score and constructive feedback. We will incorporate all of the identified limitations, suggested future work, and other changes into the final version of the manuscript. Thank you for your time and guidance.

---

### Official Review · Reviewer_bYie · 2025-07-03

**Clarity:** 3
**Significance:** 3
**Originality:** 3
**Rating:** 4
**Confidence:** 2

**Summary:**

This paper proposes *recourse loss* which measures how easily a prediction can be flipped with a low-cost, feasible action. This paper is the first to extend recourse guarantees to GBDTs: the training objective combines standard predictive loss and recourse loss via a trade-off parameter. The recourse loss is non-differentiable, so the authors derive a smooth upper bound using techniques like log-sum-exp, enabling gradient boosting.

Post-hoc refinement treats the tree ensemble as a linear model in its leaves and solves a constrained optimization to reduce recourse loss on unseen data. The optimization is made feasible by being reformulated into a tractable unconstrained problem using a Lagrangian multiplier tuned on a calibration set.

**Questions:**

See Weaknesses

**Ethical Concerns:**

["NO or VERY MINOR ethics concerns only"]

**Final Justification:**

The author have adequately addressed my concern. I believe this is a quality paper to a somewhat limited audience.

**Limitations:**

Yes

**Quality:**

3

**Strengths And Weaknesses:**

Pros:

The paper is generally well-written and the empirical result is strong. The contribution appear to be significant within a community: previous recourse-aware learning methods targeted NN which rely on differentiability, or classification trees. This paper is the first to extend recourse guarantees to GBDTs.

Cons:

There isn't an ablation study on the post-hoc refinement. Does the proposed method not make sense at all without the refinement?

---

> ### Author Rebuttal · Authors · 2025-07-31
>
> We would like to thank the reviewer for valuable and thoughtful feedback. We will reflect all of them in our final version. In the following, we will respond to the key comment raised by the reviewer.
>
> ---
> > Cons:
> > There isn't an ablation study on the post-hoc refinement. Does the proposed method not make sense at all without the refinement?
>
> Thank you for your important comment. As you mentioned, our current manuscript does not explicitly show ablation studies on our post-hoc refinement. However, we can discuss its effect in an ablation study manner by reorganizing the results in Figures 1 and 4. In what follows, we show the reorganized results from Figure 1 and Figure 4 to compare the performances of *Vanilla*, *RABIT*, *Vanilla w/ Refinement*, and *RABIT w/ Refinement*.
>
> Note that there are two main evaluation metrics (i.e., accuracy and recourse ratio) and the parameter $\lambda$ of our post-hoc refinement affects these metrics. To make the comparison easier to understand, for each method and dataset, we selected the value of $\lambda$ that attained the closest accuracy to that of RABIT. The following Table 1 shows the average accuracy of each method (higher is better). We can see that the accuracy gaps between methods were at most 2.7%, 0.6%, 0.3%, and 1.0% on FICO, COMPAS, Adult, and Bail datasets, respectively. It indicates that *all the methods attained comparable accuracy in each dataset*.
>
> **Table 1. Average Accuracy (higher is better)**
> |                       |   FICO |   COMPAS |   Adult |   Bail |
> |:----------------------|-------:|---------:|--------:|-------:|
> | Vanilla               |  0.735 |    0.682 |   0.852 |  0.711 |
> | RABIT                 |  0.732 |    0.677 |   0.851 |  0.701 |
> | Vanilla w/ Refinement |  0.708 |    0.676 |   0.853 |  0.702 |
> | RABIT w/ Refinement   |  0.716 |    0.677 |   0.850 |  0.706 |
>
> The following Table 2 shows the average recourse ratios of each method (higher is better). We observed that RABIT w/ Refinement performed the best, and that RABIT and Vanilla w/ Refinement performed better than Vanilla. In addition, we also observed that while RABIT outperformed Vanilla w/ Refinement in FICO and COMPAS datasets, Vanilla w/ Refinement outperformed RABIT in Adult and Bail datasets. These results indicate that *our learning algorithm and post-hoc refinement are effective individually*, and *their combination can achieve better results*.
>
> **Table 2. Average Recourse Ratio (higher is better)**
> |                       |   FICO |   COMPAS |   Adult |   Bail |
> |:----------------------|-------:|---------:|--------:|-------:|
> | Vanilla               |  0.541 |    0.832 |   0.274 |  0.364 |
> | RABIT                 |  *0.825* |    *0.936* |   0.427 |  0.628 |
> | Vanilla w/ Refinement |  0.694 |    0.887 |   *0.685* |  *0.840* |
> | RABIT w/ Refinement   |  **0.859** |    **0.986** |   **0.847** |  **0.868** |
>
>
> In summary, we confirmed that our proposed learning method can make sense on its own, even without our refinement method. In our final version, we will add the above ablation study and discuss its results.
>
>
> ---
> We hope that we have adequately addressed all your questions and concerns. Please let us know if we can provide any further details and/or clarifications. Thank you again for your valuable feedback.

---

### Official Review · Reviewer_H9PE · 2025-07-05

**Clarity:** 3
**Significance:** 3
**Originality:** 3
**Rating:** 4
**Confidence:** 4

**Summary:**

The paper proposes a new framework, Recourse-Aware gradient Boosted decIsion Trees (RABIT) that learns tree ensemble models with recourse guarantees. The authors further proposes a post-processing procedure for modifying an existing tree ensemble model for better recourse outcomes on unseen data. They run a empirical experiments to verify that the resulting models have recourse and do not significantly sacrifice predictive accuracy.

**Questions:**

- Why do we assume that $l_{01} (y,\hat{y}) \leq l(y,\hat{y})$? (Line 80)
- Shouldn't the 0-1 loss be $l_{01}(y,\hat{y} = \mathbb{I}[y\cdot\hat{y} < 0])$?
- Can the authors elaborate on how each term of $\xi_{t}$ can be computed in $\mathcal{O}(I_{s})$? Is this because $\mathcal{A}$ is an $l_{\infty}$ ball around $x$?
- Is Proposition 5 really necessary?
- The current framework aims to have one point with the desired prediction for each data point. Can we modify the framework to allow more recourse provision for each data point (i.e., rather than having just one with minimal cost)?
- Any guesses on when post-processing is as effective as RABIT ensembles for standard tree ensembles?

**Ethical Concerns:**

["NO or VERY MINOR ethics concerns only"]

**Final Justification:**

The authors have adequately addressed most of my concerns.

**Limitations:**

Yes

**Quality:**

2

**Strengths And Weaknesses:**

## Strengths
The authors have framed the problem very well. The problem they aim to address is significant and seems to be up to date with the literature. Experiments are well-designed.

## Weaknesses
**Significance of Experimental Results**

The paper can benefit from a more extensive discussion of the experimental results rather than just describing the outcome. The authors should take the extra step and answer: what significance do these results have? (i.e., real-life impact)

**Set of Feasible Actions**

Studies on algorithmic recourse heavily depend on how the set of feasible actions is defined. The set that the authors have presented in Section 2.1 (lines 94-95) is a convex relaxation of the true set of feasible actions. I have my reservations about this setup in effectively recovering recourse actions that are feasible in practice.

---

> ### Author Rebuttal · Authors · 2025-07-31
>
> We would like to thank the reviewer for valuable and thoughtful feedback. We will reflect all of them in our final version. In the following, we will respond to the key comments and questions raised by the reviewer.
>
> ---
> > Significance of Experimental Results
> > The paper can benefit from a more extensive discussion of the experimental results rather than just describing the outcome. The authors should take the extra step and answer: what significance do these results have? (i.e., real-life impact)
>
> We appreciate the reviewer's suggestion to elaborate on the real-life impact of our experimental results. We believe that the significance of our findings, particularly the improved recourse ratio without significant degradation in accuracy and computational efficiency, lies in their direct implications for reliability in high-stakes decision-making systems.
>
> In real-world applications like loan approvals, our RABIT provides more individuals with feasible recourse actions, as shown in Figure 1. This means that users who receive an unfavorable decision are more likely to be given clear, low-cost steps they can take to change the outcome. This directly addresses the critical need for transparency in automated decision systems. Furthermore, as shown in Figures 1 and 2, RABIT maintains comparable accuracy and computational efficiency to standard gradient boosting. This fact makes our RABIT a highly practical tool for real-world deployment. Decision-makers can adopt our method without sacrificing predictive performance or incurring prohibitive computational costs.
>
> Our results also have another real-life impact in the sense that we focus on gradient boosted decision trees (GBDTs). GBDTs, such as XGBoost and LightGBM, are one of the most popular machine learning models and are widely used due to their excellent performance. Our results indicate the potential of our method that practitioners can replace the model in their system with our RABIT and improve their ability to provide recourses without degrading accuracy and efficiency.
>
> In summary, our experimental results have significance in the sense that our RABIT is demonstrated to provide more individuals with feasible recourse actions without sacrificing predictive performance and computational efficiency of GBDTs. We believe it has a real-life impact as a first step to improve the reliability in high-stakes automated decision-making systems from the perspective of algorithmic recourse.
>
>
> > Set of Feasible Actions
> > Studies on algorithmic recourse heavily depend on how the set of feasible actions is defined. The set that the authors have presented in Section 2.1 (lines 94-95) is a convex relaxation of the true set of feasible actions. I have my reservations about this setup in effectively recovering recourse actions that are feasible in practice.
>
> Thank you for your important comment. As you pointed out, we consider a convex set defined by $\mathcal{A}(x) = [l_1, u_1] \times \dots \times [l_D, u_D]$ as a set of feasible actions, and this setup reflects the practical feasibility of actions to some extent. As mentioned in lines 95-97, this setup can express some basic constraints on actionability, such as immutable features (e.g., race) and partially actionable features (e.g., education level), by adequately setting the bounds $l_d$ and $u_d$. Thus, we believe that our method is valuable as a foundational step for further extension.
>
> Of course, we acknowledge that a more complex, non-convex set of feasible actions might better capture certain real-world nuances (e.g., dependencies between features). Unfortunately, it is not trivial to incorporate such a non-convex action set into our learning algorithm while maintaining computational efficiency and theoretical guarantees. This is because our algorithm relies on the property that we can evaluate the feasibility of an action by checking each feature independently.
>
> As a side note, we would like to note that while it is difficult for our learning algorithm to directly handle a non-convex feasible action set, our post-processing method can deal with such a complex constraint. Our post-processing method assumes only that we have an oracle algorithm $A_\beta^\ast$ for generating recourse actions, and does not require its feasible action set to be convex. Thus, if we can employ an oracle algorithm $A_\beta^\ast$ that can handle non-convex constraints (e.g., integer optimization-based algorithms), we can take into account a non-convex set of feasible actions by combining such an oracle $A_\beta^\ast$ and our post-processing method. In the final version, we will clarify this point.
>
>
> > Why do we assume that $l_{01}(y, \hat{y}) \leq l(y, \hat{y})$? (Line 80)
> > Shouldn't the 0-1 loss be $l_{01}(y, \hat{y}) = \mathbb{I}[y \cdot \hat{y} < 0]$?
>
> We appreciate your careful reading. As you pointed out, $l_{01}(y, \hat{y}) = \mathbb{I}[y \cdot \hat{y} < 0]$ is correct. We need the assumption $l_{01}(y, \hat{y}) \leq l(y, \hat{y})$ only for proving our PAC-style bound in Proposition 3. In practice, however, this assumption does not affect our learning algorithm itself, and thus our method works without it. We will clarify this point in our final version.
>
>
> > Can the authors elaborate on how each term of $\xi_t$ can be computed in $\mathcal{O}(I_s)$? Is this because $\mathcal{A}$ is an $l_\infty$ ball around $x$?
>
> Yes, we can compute each term of $\xi_t$ in $\mathcal{O}(I_s)$ by using our assumption on the $\ell_\infty$-type cost function. To compute each term of $\xi_t$, we need to enumerate the leaves of a tree $f_s$ that an instance $x$ can reach within the given cost budget $\beta$. We can identify such leaves by traversing the tree $f_s$ from its root and evaluating whether $x$ can reach the region corresponding to each node by some action whose cost is less than $\beta$. This evaluation can be done in a constant time because we need to check only the feature corresponding to the branching rule of each node, thanks to our assumption on the cost function. The entire procedure can be done by simple DFS or BFS over the tree, and takes at most the total number of nodes of $f_s$, i.e., $\mathcal{O}(I_s)$.
>
>
> > Is Proposition 5 really necessary?
>
> We would like to note that our paper does not include Proposition 5. Thus, we reply to this comment by assuming that it refers to any one of Propositions 1-3 in our paper. First, Proposition 1 gives a theoretical foundation for deriving a differentiable upper bound on recourse loss. Proposition 2 proves that taking into account the recourse loss does not sacrifice the computational efficiency of the standard gradient boosting algorithm. Finally, Proposition 3 provides a probabilistic guarantee for the existence of actions for unseen test instances. Since these implications are strongly connected to our contributions, we believe that Propositions 1-3 are necessary for guaranteeing and demonstrating the efficacy of our framework.
>
>
> > The current framework aims to have one point with the desired prediction for each data point. Can we modify the framework to allow more recourse provision for each data point (i.e., rather than having just one with minimal cost)?
>
> Thank you for your interesting suggestion. As you mentioned, our recourse loss encourages the model to guarantee the existence of at least one valid action whose cost is less than a given budget. It is not trivial to extend our recourse loss to take multiple actions into account without harming its tractability. However, we believe that our post-processing approach can be extended to allow multiple actions. If we assume that the oracle algorithm $A^\ast_\beta(x)$ generates multiple actions for $x$, we can easily extend our leaf refinement problem in (9) to the case where each instance $x_n$ has multiple actions, which can be solved by the same algorithm as our current approach. By solving this extended task, a refined model is expected to probably ensure that multiple generated actions are valid for each $x$. That is, the refined model probably guarantees the existence of multiple valid recourse actions for each instance.
>
>
> > Any guesses on when post-processing is as effective as RABIT ensembles for standard tree ensembles?
>
> Thank you for your important question. Recall our post-processing method refines the leaf weights of a trained ensemble, while fixing the tree structures in the ensemble. Thus, if an ensemble trained by the standard algorithm (Vanilla) has similar tree structures to those of an ensemble that attains a high recourse ratio, we can achieve a comparable recourse ratio to RABIT by our post-processing method alone.
>
> In our reply to Reviewer bYie, we show our ablation results where we compared the performance of Vanilla, RABIT, Vanilla w/ Refinement, and RABIT w/ Refinement. We observed that while RABIT outperformed Vanilla w/ Refinement in FICO and COMPAS datasets, Vanilla w/ Refinement outperformed RABIT in Adult and Bail datasets. This suggests that, among these four datasets, Adult and Bail are expected to be the cases where our post-processing is more effective than RABIT. While such a gap might be caused by some specific properties of datasets, such as the ratio of actionable features or the percentage of categorical features, it is challenging to characterize a general cause. It is important for future work, and we will add a discussion on this point to our final version.
>
>
> ---
> We hope that we have adequately addressed all your questions and concerns. Please let us know if we can provide any further details and/or clarifications. Thank you again for your valuable feedback.

---

> > ### Comment · Reviewer_H9PE · 2025-08-03
> >
> > I thank the authors for their detailed response. They have addressed most of my concerns and am happy to raise my score.
> >
> > I recommend the authors add what they have written in the response in regards to the significance of the empirical results.
> >
> > Overall, I believe this is a strong paper, especially after the clarification on how the post-processing method can adapt to more complex actionability constraints. I would strongly suggest the authors investigate this further and perhaps add a demonstration in the main body or the appendix.
> >
> > P.S. My question about proposition 5 was for another paper. Apologies!

---

> > > ### Author Response · Authors · 2025-08-09
> > >
> > > We thank you for your positive and constructive feedback. We are very happy to hear that our response has addressed most of your concerns and that you now consider our paper to be strong.
> > >
> > > As suggested, we will incorporate the clarification regarding the significance of our empirical results, as well as the demonstration of the adaptivity of our post-processing method, into our final version. We are confident that these changes will significantly improve our paper.
> > >
> > > We thank you again for your time and valuable insights.

---

### Official Review · Reviewer_FRpp · 2025-07-05

**Clarity:** 4
**Significance:** 3
**Originality:** 3
**Rating:** 4
**Confidence:** 4

**Summary:**

The paper studies how to train boosted tree ensembles to maximize both the predictive accuracy and the likelihood of obtaining counterfactual explanations. It shows how to adapt the training procedure and provides theoretical results on guarantees of finding feasible recourse actions.

**Questions:**

- What part specifically in the approach of Kanamori et al. [ 28 ] makes it impossible to adapt the method to gradient-boosted trees?

**Ethical Concerns:**

["NO or VERY MINOR ethics concerns only"]

**Limitations:**

See weaknesses.

**Paper Formatting Concerns:**

None.

**Quality:**

3

**Strengths And Weaknesses:**

The paper is very well written. Its main argument is convincing and the methodology is well-supported.

My main concern is w.r.t the choice of Feature Tweaking as an oracle for finding an algorithmic recourse in the experiments. This algorithm is well-known to be outperformed by several more recent approaches, such as the work of Kanamori et al, Cui et al (cited in the paper) but also:
- Parmentier, A., & Vidal, T. (2021, July). Optimal counterfactual explanations in tree ensembles. In International conference on machine learning (pp. 8422-8431). PMLR.
- Lucic, A., Oosterhuis, H., Haned, H., & de Rijke, M. (2022, June). FOCUS: Flexible optimizable counterfactual explanations for tree ensembles. In Proceedings of the AAAI conference on artificial intelligence (Vol. 36, No. 5, pp. 5313-5322).
It is likely that the trend in results would maintain with alternative algorithms for algorithmic recourse, but it is a worthy experiment to show how large the gap in guaranteed recourse depends on the underlying algorithm.

---

> ### Author Rebuttal · Authors · 2025-07-31
>
> We would like to thank the reviewer for valuable and thoughtful feedback. We will reflect all of them in our final version. In the following, we will respond to the key comments and questions raised by the reviewer.
>
> ---
> > My main concern is w.r.t the choice of Feature Tweaking as an oracle for finding an algorithmic recourse in the experiments. This algorithm is well-known to be outperformed by several more recent approaches, such as the work of Kanamori et al, Cui et al (cited in the paper) but also:
> >
> > - Parmentier, A., & Vidal, T. (2021, July). Optimal counterfactual explanations in tree ensembles. In International conference on machine learning (pp. 8422-8431). PMLR.
> > - Lucic, A., Oosterhuis, H., Haned, H., & de Rijke, M. (2022, June). FOCUS: Flexible optimizable counterfactual explanations for tree ensembles. In Proceedings of the AAAI conference on artificial intelligence (Vol. 36, No. 5, pp. 5313-5322).
> >
> > It is likely that the trend in results would maintain with alternative algorithms for algorithmic recourse, but it is a worthy experiment to show how large the gap in guaranteed recourse depends on the underlying algorithm.
>
> We appreciate your important suggestion. While we employed the Feature Tweaking algorithm due to its efficiency in our experiments, we agree with the worthiness of the experiments that examine the dependence on the underlying algorithm for extracting recourse actions.
>
> We conducted additional experiments where we employed the exact algorithm based on integer optimization proposed by (Cui et al. 2015), which is one of the alternative algorithms you raised. As with the experiments in Section 5.1, we split each dataset into the training and test sets with a ratio of 75:25, and trained a tree ensemble model by each method on the training set. Then, for test instances, we extracted actions from the model by (Cui et al. 2015), and measured the average cost of the extracted actions. Due to computational costs, we randomly sample 50 test instances for each trial.
>
> The following Table 1 presents the average cost of actions over 10 trials (lower is better). We can see that our RABIT achieved lower costs than the baselines, regardless of the datasets, showing a similar tendency to the results with the Feature Tweaking algorithm. These results suggest that *the effectiveness of our method is maintained with an alternative recourse algorithm based on integer optimization*.
>
> **Table 1. Average Cost of Actions Extracted by (Cui et al. 2015) (lower is better)**
> |         |   FICO |   COMPAS |   Adult |   Bail |
> |:--------|-------:|---------:|--------:|-------:|
> | Vanilla |  0.354 |    0.188 |   0.340 |  0.420 |
> | OAF     |  0.293 |    0.158 |   0.297 |  0.320 |
> | RABIT   |  **0.110** |    **0.096** |   **0.257** |  **0.213** |
>
> We will add these results to our final version. If possible, we will also conduct additional experiments with (Parmentier & Vidal 2021) or (Lucic et al. 2022).
>
>
> > What part specifically in the approach of Kanamori et al. [ 28 ] makes it impossible to adapt the method to gradient-boosted trees?
>
> Thank you for your important question. The learning algorithm of Kanamori et al. is designed for classification trees and implicitly exploits the assumption that the output space is a finite set of labels, such as $\{ -1, +1 \}$. It enables us to efficiently determine the optimal leaf labels for a fixed branching rule by enumerating all the patterns of leaf labels and computing the objective value of each pattern. However, gradient boosting requires learning regression trees, where we need to handle the output space over $\mathbb{R}$, rather than a finite set of labels. This prevents us from directly applying the existing algorithm of Kanamori et al. to our learning problem, and thus, we develop a new learning algorithm designed for gradient boosting.
>
> ---
> We hope that we have adequately addressed all your questions and concerns. Please let us know if we can provide any further details and/or clarifications. Thank you again for your valuable feedback.

---

### Note · Authors · 2025-08-14

We are grateful to the reviewers and the AC for their thoughtful feedback and the opportunity to improve our work. We were particularly pleased that *the reviewers acknowledged our writing quality (FRpp, bYie), the significance of our problem (H9PE, JBHL, nKF5), and the theoretical and empirical support for our methodology (FRpp, H9PE, bYie, JBHL)*. The feedback led to crucial clarifications and improvements in our manuscript.

To address reviewer **FRpp**'s concern about our choice of recourse algorithms, we conducted additional experiments with another algorithm (Cui et al., 2015). We confirmed our method remains effective, and we believe this result has at least partially addressed the reviewer's concern.

For reviewer **H9PE**, we clarified the practical importance of our method in high-stakes decision-making systems. We also explained our assumptions on feasible actions and how our post-processing method adapts to more complex actionability constraints. *The reviewer explicitly acknowledged that most of the concerns were addressed and promised to increase the score*.

To address reviewer **bYie**'s concern about the lack of ablation studies, we reorganized the results in Figures 1 and 4 to discuss the effect of our post-hoc refinement in an ablation study manner. We demonstrated that both our learning algorithm and our post-hoc refinement are effective on their own and that combining them yields the best results. We believe we have fully addressed the reviewer's concern.

To answer reviewer **JBHL**, we clarified the practicality of our approach by explaining how we incorporate actionability constraints during both training and recourse generation to ensure realistic actions. We also provided more detailed explanations of our experimental observations, such as the performance of OAF and the impact of $\gamma$. *Following the rebuttal, the reviewer promised to retain the positive score*.

Finally, for reviewer **nKF5**, we addressed several points, including the rationale behind our dataset choices and the applicability of our method to multiclass settings. We also clarified our statement about the trade-off between recourse ratio and accuracy. *The reviewer also promised to keep the positive score after the rebuttal*.

We are committed to incorporating all discussed improvements into our final version. We would like to once again thank all the reviewers and the AC for their constructive and insightful feedback.

---

### Decision · Program_Chairs · 2025-09-17

**Decision:**

Accept (poster)

**Comment:**

This paper proposes a gradient boosting algorithm for the _algorithmic recourse_ framework, in which the predictor can provide a _recourse action_ that would have changed its prediction. For example, if the predictor outputs a credit score, it could suggest a change to the candidate's financial status that would have increased (or decreased) their score. This helps with explainability and could be used to improve outcomes (e.g., suggesting actions a candidate could take to improve their credit). According to the paper, guaranteeing the existence of a recourse action is nontrivial, and prior methods to guarantee recourse actions do not apply to the class of gradient boosting algorithms. Thus, the paper fills this gap, providing a gradient boosting algorithm with (high-probability) recourse guarantees. In addition, it provides a post-processing algorithm that, given a learn model, outputs a "refined" version thereof along with recourse guarantees.

The reviews all say that this paper is well written, well motivated, and technically sound. The authors' rebuttal seems to have addressed all of the reviewers' concerns, so I will not bother enumerating them here. The overall scores are all positive, so I recommend accepting this paper.